# Large-volume focus control at 10 MHz refresh rate via fast line-scanning amplitude-encoded scattering-assisted holography

Atsushi Shibukawa[1], Ryota Higuchi[1], Gookho Song [2], Hideharu Mikami [1] ✉, Yuki Sudo [3] ✉ & Mooseok Jang [2] ✉

The capability of focus control has been central to optical technologies that require both high temporal and spatial resolutions. However, existing varifocal lens schemes are commonly limited to the response time on the microsecond timescale and share the fundamental trade-off between the response time and the tuning power. Here, we propose an ultrafast holographic focusing method enabled by translating the speed of a fast 1D beam scanner into the speed of the complex wavefront modulation of a relatively slow 2D spatial light modulator. Using a pair of a digital micromirror device and a resonant scanner, we demonstrate an unprecedented refresh rate of focus control of 31 MHz, which is more than 1,000 times faster than the switching rate of a digital micromirror device. We also show that multiple micrometer-sized focal spots can be independently addressed in a range of over 1 MHz within a large volume of 5 mm × 5 mm × 5.5 mm, validating the superior spatiotemporal characteristics of the proposed technique – high temporal and spatial precision, high tuning power, and random accessibility in a three-dimensional space. The demonstrated scheme offers a new route towards three-dimensional light manipulation in the 100 MHz regime.

High-speed and precision optical focus control has long served as a basis to construct optical systems with high temporal and spatial resolutions. 3D laser-scanning microscopy[1,2] and laser micromachining[3] are prominent examples where a high spatio-temporal resolution plays a critical role when observing fast sub-cellular dynamics and achieving high-throughput material processing. Traditionally, 3D focus control has been implemented with a 2D beam scanner (e.g., galvanometer scanner[1,2], acousto-optic deflector (AOD)[4]) and an objective lens driven by a piezo actuator[1,2,5]. In this traditional scheme, the speed of axial focus control is typically much slower (typically < 1 kHz) than that of transversal control (typically ~1 MHz), as it involves precise mechanical positioning of bulky focusing optics.

Many varifocal lens schemes have been proposed to overcome this speed discrepancy[6]. Their working principles are broadly categorized into mechanical, electro-optic (EO), and acousto-optic (AO) modulation. In the mechanical type, the shapes of varifocal elements are directly modulated through electrostatic[7,8] or mechanical forces[9–11]; however, due to the necessity of precise physical positioning, the operation speed is typically limited to 10 kHz. In contrast, EO[12]- and AO[13,14]-based approaches directly modulate the refractive index profile of a fixed medium via electric or acoustic field, achieving control speeds of up to 1 MHz. However, considering the fundamental limiting factors, i.e., the finite capacitance of EO ceramics and the speed of sound in AO materials, realizing operation speeds beyond 1 MHz is highly challenging. Moreover, the axial scan power (i.e., tuning power) is restricted to ~10 m⁻¹ because the materials' responses to applied fields, characterized as the Kerr coefficient or piezo-optic coefficient, are often very small[6], thereby limiting the axial scan range

[1]Research Institute for Electronic Science, Hokkaido University, Sapporo 001-0020, Japan. [2]Department of Bio and Brain Engineering, Korea Advanced Institute of Science and Technology, Daejeon 34141, Republic of Korea. [3]Faculty of Medicine, Dentistry and Pharmaceutical Sciences, Okayama University, Okayama 700-8530, Japan. ✉e-mail: hmikami@es.hokudai.ac.jp; sudo@okayama-u.ac.jp; mooseok@kaist.ac.kr

in 3D multiphoton microscopy[13,15] and laser micromachining[14]. Furthermore, the effects of the speed-limiting factors become more pronounced with the varifocal elements of larger apertures, resulting in the fundamental trade-off between the response time and the tuning power.

Another route towards active focus control is the use of programmable spatial light modulators (SLM), (e.g., liquid crystal on silicon; LCoS[16,17], digital micromirror device; DMD[18,19], and deformable mirror; DM[20]). The ability to arbitrarily reconfigure a complex 2D wavefront allows not only transverse control but also axial control in a random-access fashion, unlike conventional varifocal elements. The random-access capability allows for the selective illumination of desired points in a 3D space, leading to, for example, reduced photodamage to a given specimen in biological microscopy[15,21] and high-throughput laser micromachining[22]. Unfortunately, despite these capabilities, the tuning power of SLMs is even worse to less than 10 m$^{-1}$ due to their limited spatial degrees of freedom (DOF) (i.e., the number of independently controllable pixels). We note that, recently, the method of utilizing multiple AODs modulated with chirped sinusoidal signals[13,15,21] has also been widely used in fast 3D random-access scanning, but its tuning power is limited to the similar value as for the SLMs[6].

Interestingly, in conjunction with a scattering medium, SLM-based approaches have been shown to provide superior spatial characteristics—high-NA focusing and an extremely broad 3D scan range—regardless of the magnitude of their intrinsic spatial DOF[23,24]. However, the focus control speed is inherently limited by their temporal DOF (i.e., the refresh rate) of conventional SLMs (e.g., LCoS and DMD), which is typically orders of magnitude lower (i.e., ranging from 10 Hz to 20 kHz) compared to those of EO- and AO-based varifocal lenses. Only recently has the use of a grating light valve (GLV) enabled 1D wavefront modulation and holographic optical focusing at 350 kHz[25]. Some novel designs for EO-based SLMs have shown response times shorter than 1 μs[26–28], albeit with a handful of spatial DOFs which is insufficient for 3D focus control.

In this work, we propose a novel ultrafast wavefront modulation method that effectively transforms a 2D-SLM into an ultrafast 1D-SLM via line-beam scanning. Through the process of reallocating the spatial DOF to the temporal DOF, our method amplifies the speed of wavefront modulation by the factor of up to a few thousands, surpassing the current speed limit of 1D-SLMs (e.g., GLV) by two orders of magnitude. Using a scattering medium as a holographic focusing element, we further develop a 3D focus control technique, termed fast line-scanning amplitude-encoded scattering-assisted holographic (FLASH) focusing, and achieve record-high speeds of up to 31 MHz for 3D focus control. Using the FLASH focusing technique, we also demonstrate random-access control of micrometer-sized focal spots over an axial range of 0.01–10 mm (i.e., tuning power of around 100,000 m$^{-1}$) and a transverse range of 5 mm × 5 mm at refresh rates higher than 10 MHz, opening up new opportunities for optical interrogation with extremely high spatial complexity and fast dynamics.

## Principle

Figure 1 represents the principle of the proposed method. First, to achieve ultrafast wavefront modulation, a resonant scanner (RS) performs transverse scanning of a line beam on a 2D-DMD through a cylindrical lens. During the oscillatory scanning, the amplitude profile of the line beam is sequentially modulated by independent binary patterns on each DMD region illuminated by the line beam. The amplitude-encoded line beam then reverses the incident path and is expanded back into a circular beam with a striped pattern such that its projected area is fixed regardless of the line beam position on the DMD (i.e., scanning position of the RS). The RS and the DMD can be replaced by any 1D beam scanner and 2D-SLM to implement the proposed high-speed wavefront modulation technique.

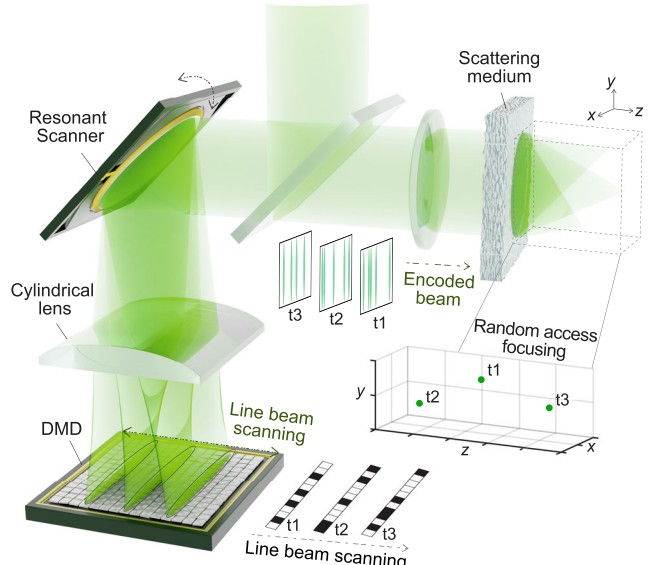

**Fig. 1 | Schematic of the FLASH focusing.** A resonant scanner and a cylindrical lens achieve fast line beam scanning across a digital micromirror device (DMD). As the line beam moves across the DMD columns, its amplitude profile is sequentially modulated with different pre-calibrated binary patterns on the DMD. The amplitude-encoded line beam is then expanded back into a 2D striped beam and projected onto a fixed area on a scattering medium regardless of the motion of the resonant scanner. Finally, as the line beam oscillates on the DMD, the time-varying amplitude-modulated beam is holographically focused onto different 3D positions beyond the scattering medium in a random-access manner.

There are four important parameters that determine the overall performance of our wavefront modulation technique: the line-scan frequency in bidirectional scanning of the beam scanner ($f_{scan}$), the number of pixels on the SLM ($N \times M$; where $N$ and $M$ denote the numbers of rows and columns along the major and minor axes of the line beam, respectively), and the number of columns illuminated with a single line beam ($M_{col}$). The wavefront of the line beam is encoded by the illuminated columns on the SLM with the spatial DOF, DOF$_{spatial}$, of $N \times M_{col}$. For a single line-scan over the SLM which is performed within $1/f_{scan}$, the line beam scans across $M/M_{col}$ independent wavefront-modulating columns, resulting in a refresh rate of the wavefront modulation of

$$f_{mod} = f_{scan} \times \kappa, \quad (1)$$

where $\kappa = \frac{M}{M_{col}}$ is the speed gain.

In general, the spatiotemporal DOFs (DOF$_{spatiotemporal}$) of wavefront modulation techniques can be dictated by the product of the number of spatially independent wavefronts (DOF$_{spatial}$) and the number of independent time points within the unit time (DOF$_{temporal}$) that can be addressed with the SLM,

$$DOF_{spatiotemporal} = DOF_{spatial} \times DOF_{temporal}. \quad (2)$$

Practically, DOF$_{spatial}$ is identical to the number of independently controllable pixels on the SLM and DOF$_{temporal}$ is identical to the intrinsic refresh rate of the SLM, $f_{SLM}$, or the number of independently controllable frequencies per unit time using the SLM. In this regard, with an assumption of $f_{scan} \geq f_{SLM}$, our method provides a simple but powerful way to implement the gain in DOF$_{temporal}$ (i.e., speed gain, $\kappa$) at the cost of DOF$_{spatial}$ under the trade-off relationship, DOF$_{spatial} \times \kappa = N \times M$, the intrinsic DOF$_{spatial}$ of the SLM. For example, if we set $M_{col}$ to 1, the speed gain $\kappa$ is maximized to $M$ while DOF$_{spatial}$ is minimized to $N$. In this setting, assuming a standard DMD with $N$ and $M$

over 1,000 and a RS with $f_{scan}$ equal to $f_{SLM}$ (typically over 10 kHz), we can expect a refresh rate (i.e., $DOF_{temporal} = \kappa \times f_{SLM}$) of over 10 MHz (i.e., $\kappa > 1,000$) with $DOF_{spatial}$ of ~1000. We note that various techniques have been previously developed to extend a view angle of a 3D holographic display system based on the similar components, such as 2D SLMs, cylindrical lens, and actuators[29,30]. In contrary to the FLASH technique, those existing approaches have been proposed to increase $DOF_{spatial}$ at the expense of $DOF_{temporal}$.

Another important feature of the proposed method is its capability to flexibly reallocate $DOF_{spatiotemporal}$ by tuning $M_{col}$ to address a wide range of $DOF_{temporal}$ and $DOF_{spatial}$. Ideally, standard LCoS and DMD provide the $DOF_{spatiotemporal}$ of ~$10^8$ and ~$10^{10}$ from Eq. (2) respectively. Therefore, one may purposefully set $M_{col}$ or $f_{scan}$ to address a spatiotemporal domain within the range where $DOF_{temporal} > 10^6$ and $DOF_{spatial} > 10^2$, which cannot be easily addressed with existing wavefront modulation techniques (see Supplementary Fig. 1 for the comparison).

The FLASH focusing technique is implemented using the proposed wavefront modulation method for random-access holographic focus control through a scattering medium. To create scattering-assisted focal spots, the wavefront solutions can be pre-calibrated using previously developed wavefront shaping techniques that are based on iterative methods[31,32], optical phase conjugation[33], or the transmission matrix (TM) formalism[34,35]. As every $M_{col}$ columns of the SLM can be optimized for different focal positions (i.e., $\frac{M}{M_{col}}$ focal spots can be addressed within the single line-scan), the rate of focus control, $f_{spot}$, is equal to that of wavefront modulation $f_{mod}$ in Eq. (1). As holographic focusing involves a process of constructive interference of many speckle fields contributed by different spatial input modes[36], the focal contrast $\eta$, quantified as the peak-to-background intensity ratio, is proportional to $DOF_{spatial}$, as follows:

$$\eta = \beta \times DOF_{spatial}, \qquad (3)$$

where $\beta$ is a focusing fidelity factor that depends on the type of wavefront modulation (e.g., $\pi/4$ for phase-only[36] and $1/(2\pi)$ for binary-amplitude modulation[37]). Therefore, similar to the trade-off in our wavefront modulation scheme, the focal contrast and the refresh rate of focus control are in a trade-off relationship.

## Results
### Validation of the FLASH focusing technique
To implement the setup for fast wavefront shaping, we used RS and DMD with $f_{scan}$ and $f_{SLM}$ both at 24 kHz. A line beam was projected onto the DMD through a 4× objective lens. A volume holographic grating was placed in front of the DMD to construct a retroreflective configuration in which the tilt angle of each DMD micromirror is offset by the first-order diffraction angle of an incident beam (see Methods and Supplementary Fig. 2a for details about the experimental setup). The active numbers of columns and rows on the DMD, $M$ and $N$, were set to 340 and 352, respectively, considering the field of view (FOV) of the objective lens (see Supplementary Text 1 for the optimal configuration of the FLASH system to increase $M$ and $N$ and Supplementary Fig. 3 for line beam width over the active DMD columns). We projected the amplitude-encoded striped beam on an opal diffusing glass with the area of 4 mm × 4 mm. We chose opal glass with a thickness of 0.45 mm among other scattering media owing to its isotropic scattering profile (see Supplementary Fig. 4 for the angular scattering profile), high stability (see Supplementary Fig. 5 for the temporal decorrelation profile), and high transmittance (~30 %).

To validate that line beam scanning effectively converts the spatial array of columns on the 2D-DMD into a temporal array of 1D binary patterns, we characterized the correlation matrix between the striped intensity patterns on the projection plane while illuminating different columns with the same periodic binary pattern (Fig. 2a, b). A 1D binary

pattern on the DMD was shown to be projected into a high-definition stripe pattern and the correlation values for every column pair were close to 1, confirming that the optical intensity response from every $M$ column was fixed regardless of the line beam position (i.e., the RS scanning position).

Although an illumination pattern was highly asymmetric, the target plane behind the scattering medium can be addressed with a spatially isotropic resolving power. In our configuration, each on-state pixel on a column is projected as a single thin stripe. This thin stripe caused a spatially elongated speckle field due to the short-range angular correlation of the medium (Fig. 2c, d; see Supplementary Fig. 6 for the short-range correlation profile). However, when multiple pixels were set in the on-state, interference between the elongated speckle fields resulted in a circularly symmetric autocorrelation function (Fig. 2e, f), serving as a basis to address the target plane holographically.

We then validated the focusing capability of the FLASH technique based on the procedure described in Fig. 2g. First, we measured the input-output response of the medium by displaying many different random binary patterns, $\mathbf{S}_{in} = \{\mathbf{s}_1, \ldots, \mathbf{s}_{N_{in}}\}$ on a single DMD column, and measuring the complex amplitude of the speckle patterns generated on the target plane, $\mathbf{U}_{out} = \{\mathbf{u}_1, \ldots, \mathbf{u}_{N_{in}}\}$ using phase-shifting digital holography with a reference beam. Next, with basis transformation using the pseudo-inverse matrix of $\mathbf{S}_{in}, \mathbf{U}_{out} \times \mathbf{S}_{in}^+$, we constructed the TM $\mathbf{K}$ of the position basis from which a binarized amplitude pattern of the phase-conjugated field for a desired focus is computed (i.e., turning on only elements whose phase is between 0 and $\pi$ in a desired column of $\mathbf{K}^*$) and displayed on a DMD column[19]. Figure 2h shows a focal spot reconstructed on the target plane of $z = 2$ mm from the medium. The focal spot presented a circular shape with a full-width at half-maximum (FWHM) of around 400 nm, which corresponds to a numerical aperture (NA) of 0.68. The measured focal contrast $\eta$ of 40 was consistent with the expected value of ~56 from Eq. (3) (i.e., $N/(2\pi)$ for $M_{col} = 1$). Lastly, with correction of the column-wise wavefront distortion, the focal contrast was shown to be preserved regardless of the line beam position, implying that every DMD column is treated identically in terms of the intensity and phase response for complex wavefront shaping (see Methods and Supplementary Fig. 7 for details).

### Spatial and temporal performances
We experimentally tested the spatial performance of the FLASH focusing technique in terms of the spot size and addressable 3D volume (Fig. 3a). When the spot was transversely scanned from $x = 0$ to 2.5 mm at a fixed target plane of $z = 1.5$ mm, the FWHM spot sizes of the reconstructed foci increased from 410 nm to 560 nm, corresponding to NA values ranging from 0.66 to 0.48 (Fig. 3b). When the spot was axially scanned from $z = 0.01$ to $z = 10$ mm along the optical axis, FLASH focusing provided effective NAs of 0.8–0.21, corresponding to FWHM spot sizes of 340–1290 nm, respectively (Fig. 3c). Note that we repeatedly measured TMs to address focal spots at different axial positions (see Supplementary Text 2 for available approaches for 3D focal scanning without multiple TM measurements).

This extremely large tuning range corresponds to a tuning power of around 100,000 m⁻¹, more than two orders of magnitude higher than those of conventional varifocal lenses. For 3D volumetric focusing capability, an effective NA larger than 0.45 was maintained over the 3D cylindrical space with a diameter of 5 mm and height of 2 mm (Supplementary Fig. 8), far exceeding the lateral FOV of ~1 mm and the depth of field for a commercial objective lens with 0.45 NA. This validates the feasibility of the FLASH focusing technique for use in application areas such as 3D laser-scanning microscopy and laser micromachining, where a large 3D scan range is critical.

Next, we experimentally tested the temporal performance of the proposed FLASH focusing technique. In our configuration with

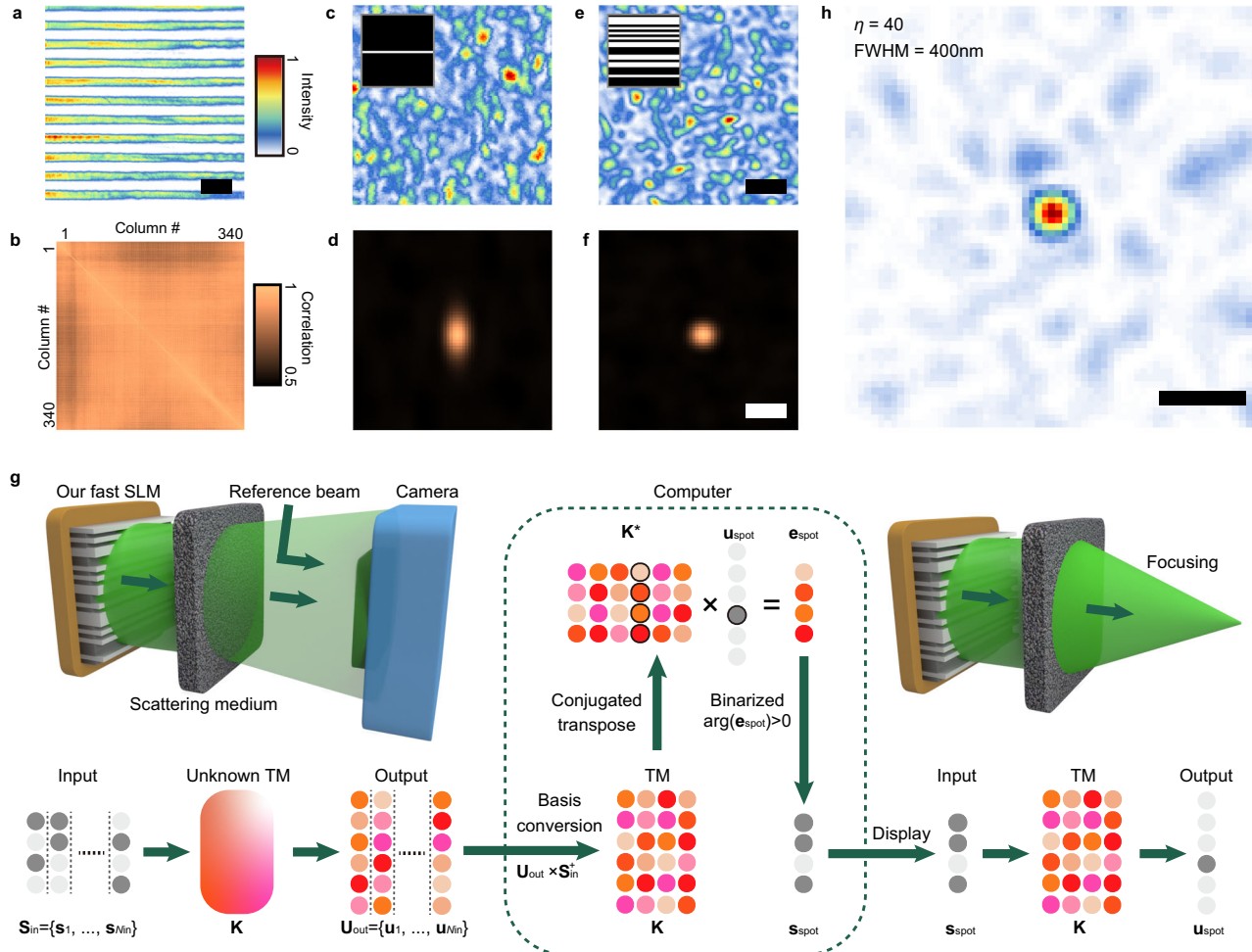

**Fig. 2 | Validation of the FLASH focusing technique. a** Intensity profile of a 2D stripe beam on the projection plane located at the input surface of a scattering medium. The illuminated DMD column was modulated with a 1D binary pattern with eight-pixel period. Scale bar: 100 μm. **b** Correlation matrix of intensity profiles between every column pair. The intensity profile projected from each DMD column was separately recorded while the same binary pattern with two-pixel period was displayed on every DMD column. **c**, **e** Speckle intensity distributions on the target plane of $z = 2$ mm when the binary patterns shown in the insets are respectively displayed on the DMD. Scale bar: 1 μm. **d**, **f** 2D autocorrelation function of the speckle distribution in (**c**, **e**). Scale bar: 1 μm. **g** Schematic diagram of holographic focusing through a scattering medium based on the transmission matrix (TM) approach. Here, $N_{in}$, $\mathbf{S}_{in}^{+}$ and $\mathbf{K}^{*}$ denote the number of input binary patterns for the TM measurement, the pseudo-inverse matrix of $\mathbf{S}_{in}$ and conjugated transpose of $\mathbf{K}$, respectively. **h** Focal spot reconstructed on $z = 2$ mm based on the measured TM of the medium.

$f_{scan} = 24$ kHz, $M = 340$, and $M_{col} = 1$, the refresh rate $f_{mod}$ is expected to be around 8 MHz based on Eq. (1), ideally for a constant scanning speed of the line beam on the DMD plane, and the line-scan range exactly matched the active DMD area of $M = 340$. However, as the DMD requires a finite time of around 26 μs to update the 2D binary patterns on the active area, we set the scanning angle range of the RS to be ~2.7 times larger than the ideal range to ensure necessary time for the continuous updating of the binary patterns on the DMD with every directional change of line beam scanning. Also, considering the sinusoidal profile of the RS's scan speed, the maximum local refresh rate (i.e., the inverse of the time required to scan the pixel pitch of 13.7 μm) is expected to be around 30 MHz with a maximum scanning speed of the line beam of 474 m/s. With these settings, at the expense of an improvement in the local refresh rate and continuous modulation capability, the duty cycle $D$, defined as the percentage of the period for active wavefront modulation relative to the total period of the single line-scan, expectedly decreases to ~24%. However, it should be noted that $DOF_{temporal}$ is given as a fixed value of ~8 MHz from Eq. (1) regardless of the value of the duty cycle or the angle scan range of the RS, considering that the fundamental $DOF_{temporal}$ can be determined as

the product of the duty cycle and the averaged local refresh rate during active modulation.

Figure 3e shows the on/off modulation signal acquired with a photomultiplier tube (PMT) during line beam scanning over DMD columns alternatively encoding optimized and random binary patterns (Fig. 3d). The contrast of on/off signals lower than the theoretical value from Eq. (3) and the considerable fluctuation of the peak intensity (i.e., the signal level at the 'on' state) were observed, which can be attributed to a large collection area (i.e., pinhole area) used in conjunction with the PMT (see Methods for the detailed experimental setup) and the intrinsic statistical fluctuation in associated TM (see Supplementary Text 3 and Supplementary Fig. 9 for the detailed description and the simulation result for the intrinsic statistical fluctuation). It should be noted that, with the collection area larger than the diffraction-limited spot, many background modes (i.e., speckle granules) were collected together with the focal spot and therefore the measured signal for PMT does not reflect the actual spot contrast. From the sinusoidal fitting shown in Fig. 3f, the on/off switching time for the focal spot was measured to be 32.5 ns at the central columns, corresponding to a refresh rate of 31 MHz. For the entire period of active modulation in

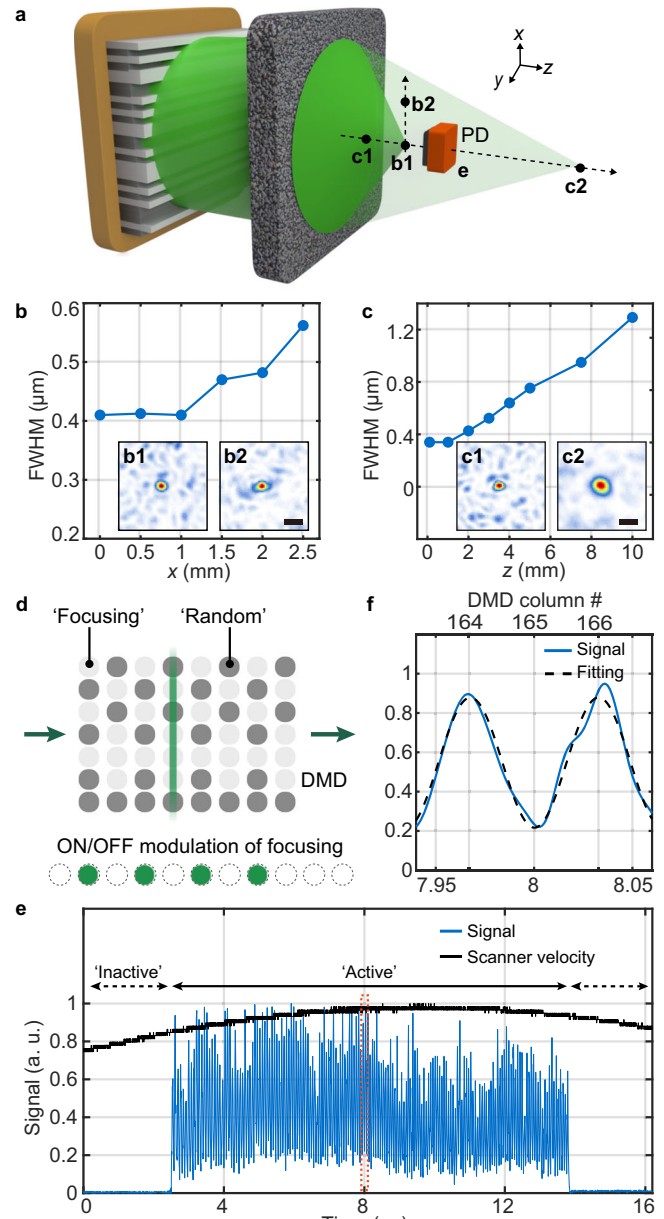

**Fig. 3 | Spatial and temporal performances of the FLASH focusing technique.** **a** Schematic of the production of focal spots over a large 3D space. A photomultiplier tube (PMT) is used for recording ultrafast modulation of the focal spot. **b** Measured full-width at half-maximum (FWHM) of foci for different $x$ positions on a plane of $z = 1.5$ mm. **b1-b2**, Measured intensity profiles of foci at $x = 0$ and 2.5 mm. Scale bar: 1 μm. **c** Measured FWHM of foci for different $z$ planes at $x = 0$ mm. **c1-c2**, Measured intensity profiles of foci at $z = 1$ and 5 mm. Scale bar: 1 μm. **d** Schematic of periodic on/off focusing modulation. The line beam is scanned across the DMD, where 'Focusing' pattern and 'Random' pattern are alternatively encoded into each single column. **e** Optical signals acquired with the PD shown in (**a**) during a single line-scan of the line beam. 'Active' time and 'Inactive' time indicate the time for serial wavefront modulation using 340 columns and the time for the update of the DMD frame, respectively. The velocity signal from the scanner board is plotted for reference in black line. **f** Close-up of the signals in the dotted orange box shown in (**e**) with the sinusoidal curve fitting in dotted black line.

Fig. 3e, the local refresh rate ranged from 25 to 31 MHz due to the sinusoidal profile of the RS scan speed. This value of the refresh rate (or response time) testifies an improvement of at least one or two orders of magnitude over state-of-the-art varifocal elements[12,14] or wavefront shaping techniques for optical focusing[25]. Moreover, with

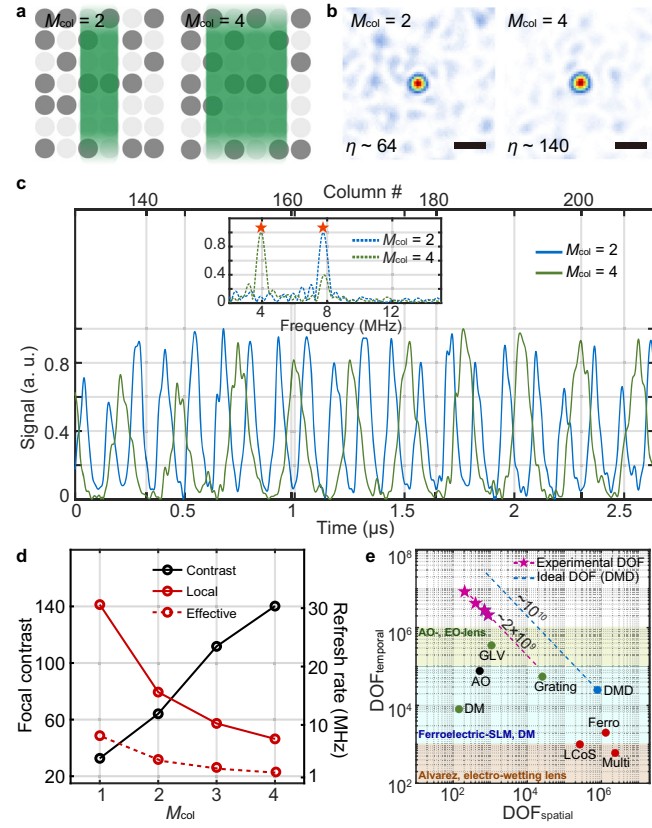

**Fig. 4 | Control of spatial and temporal degrees of freedom for wavefront modulation.** **a** Illustration of how a line beam is illuminated on a DMD with the setting of $M_{col} = 2$ and 4. **b** Intensity profiles of focal spots for $M_{col} = 2$ and 4, respectively. Scale bar: 1 μm. **c** Optical signals for the periodic on/off modulation of the focal spot for $M_{col} = 2$ and 4, respectively. The inset plot indicates the frequency spectrums of the modulated signals around the central column. **d** Contrast and refresh rate of focal spots for different values of $M_{col}$. The effective refresh rate was calculated from the product of the experimental duty cycle and the averaged local refresh rate. **e** Flexible reallocation of $DOF_{spatial}$ and $DOF_{temporal}$ by tuning $M_{col}$ in our wavefront modulation method. Pink stars indicate the experimentally demonstrated DOFs for $M_{col} = 1$–4. Dotted blue lines indicate ideal addressable DOFs using our methods based on DMD. Orange, blue, and green filled areas denote three representative categories of conventional varifocal lenses in terms of $DOF_{temporal}$ (i.e., the response speed) (see details in Ref. [6]).

the angle scan range purposefully set such that it exceeds the active DMD area, the binary patterns on the DMD could be synchronously refreshed during any directional change of line beam scanning. Accordingly, the number of independently addressable foci was not limited to the number of active columns $M$ on the DMD (Supplementary Fig. 10 and see Continuous Ultrafast Spatial Modulation in the Methods section).

## Control of spatial and temporal degrees of freedom

We demonstrated the tunability of the FLASH focusing technique in setting the focal contrast and the refresh rate under the trade-off relationship in Eq. (2) by simply varying the number of illuminated columns, $M_{col}$ (see Methods for details). Figure 4a–c present the focus control results with $M_{col} = 2$ and 4. As expected from Eqs. (1)–(3), the focal contrast increased nearly proportionally to 64 and 140 while the local on/off refresh rate around the central column correspondingly decreased to 15.6 MHz and 7.6 MHz for $M_{col} = 2$ and 4 as shown in Fig. 4d. The $DOF_{spatiotemporal}$ for different values of $M_{col}$ were experimentally characterized as the product of $\eta_{exp}/\beta$ and $f_{exp} \times D$, representing the experimental $DOF_{spatial}$ and $DOF_{temporal}$, respectively. $\eta_{exp}$ and $f_{exp}$ are the measured focal contrast and the averaged local refresh

rate. As shown in Fig. 4e, the experimental $DOF_{spatiotemporal}$ was estimated to be ~$2 \times 10^9$ regardless of $M_{col}$, which is consistent with the theoretical $DOF_{spatiotemporal}$ of $N \times M \times f_{SLM}$ at around $2.8 \times 10^9$. The discrepancy may be attributed to practical imperfections in phase conjugation process that degrade the focal contrast[38]. It is worth noting that the experimentally demonstrated DOFs (pink stars in Fig. 4e) cannot be addressed with existing modulation techniques such as the LCoS-, MEMS-, and AO-based SLM types and also the demonstrated focus control speed (i.e., $DOF_{temporal}$) exceeds those of the state-of-the-art varifocal lenses (as shown in Fig. 4e and Ref. 6). With the optimal implementation of the FLASH technique using an objective lens with lower magnification and a DMD with a higher resolution (i.e., with larger $N$ and $M$), the gap between our experimental DOF and the ideal DOF for a DMD can be even narrowed.

### Demonstration of >10 MHz random-access focusing

To demonstrate the 3D random-access capability of the FLASH focusing technique, first we demonstrated focus control over two separate planes within 1 μs. The DMD was programmed in such a way that every two columns are individually optimized to scan focal spots in elliptic patterns on the two planes of $z = 3$ mm and 3.2 mm (shown in Fig. 5a and Supplementary Fig. 2b). As the expected refresh rate of ~16 MHz (with the setting of $M_{col} = 2$) greatly exceeds the maximum frame rate of a standard camera, we instead counted the number of focal spots that can be captured over the short exposure time of 1 μs. Consistently, we observed in total 17 focal spots from the two camera images as shown in Fig. 5b, confirming the capability of 3D random-access focus control at more than 10 MHz.

Secondly, to demonstrate the versatility of the FLASH technique on the basis of the superposition principle, we performed the frequency-multiplexed modulation of two distant focal spots as shown in Fig. 5c. Specifically, we encoded the frequency information as a column period of an optimal wavefront for a certain spot location such that the modulation frequency is set to $f_{spot}$ divided by the column period. Then, we superposed and binarized the wavefronts for individual spots and locations to simultaneously address multiple frequencies and spot locations. Figure 5d, e present the temporal profiles of optical intensities and the corresponding frequency spectra as measured through two single-mode fibers placed at $(x, y, z) = (0, 2.5, 2)$ mm and $(0, 0, 2)$ mm. As the column periodicity for the two positions were respectively set to 4 and 8, the local modulation frequencies were distinctly measured to be 8 and 4 MHz, respectively. We have also demonstrated a single spot modulation over a large volume of $5\text{ mm} \times 5\text{ mm} \times 5.5\text{ mm}$ as shown in Supplementary Fig. 11. The reduced contrast of on/off signals in Fig. 5d and Supplementary Fig. 11 as compared with the expected value from Eq. (3) is due to the collection area of the fiber tip (i.e., mode-field diameter of the fiber) that is larger than the diffraction-limited size of the focal spot and the surrounding speckle granules. Those random-access focusing results indicate that the FLASH focusing technique unlocks a spatiotemporal domain that is inaccessible using conventional optics with extreme flexibility, practically enabling frequency-encoded imaging[39] or frequency-multiplexed switching over unprecedentedly large volumes and bandwidths.

### Discussion

In this study, we proposed a method to reallocate the large $DOF_{spatial}$ of the 2D-SLM to the DOF in the spatiotemporal domain in a tunable manner, thereby achieving ultrafast spatial light modulation with a refresh rate of around 10 MHz and a $DOF_{spatial}$ value exceeding 100. Our method, without a scattering medium, can serve as a general-purpose 1D-SLM such as a GLV, albeit with a much higher refresh rate. Therefore, it can be broadly used in high-definition panoramic display[40], maskless lithography[41], and spectral shaping[42] applications. Combined with scattering media, the FLASH focusing technique

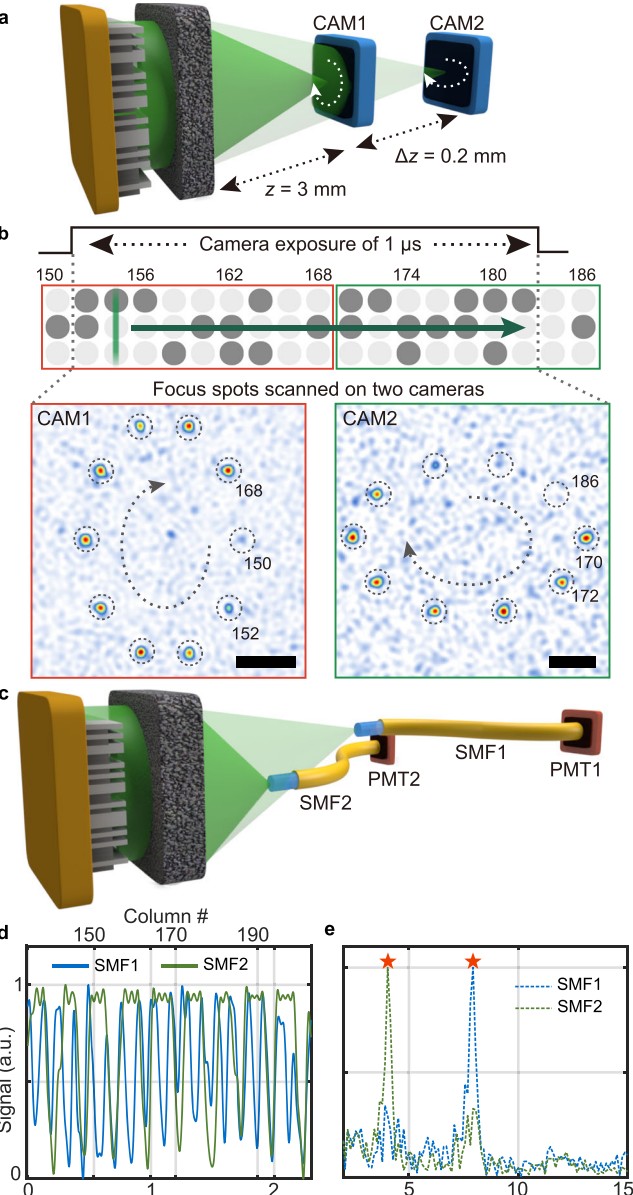

**Fig. 5 | Ultrafast random-access focus control in a large 3D space. a** Simplified experimental schematic for ultrafast 3D focus control using two cameras (CAM1 and CAM2) located on axially separated target planes of $z = 3$ mm and 3.2 mm. Within a camera exposure time of 1 μs, the line beam scans over ~30 columns and undergoes serial wavefront modulation in the setting of $M_{col} = 2$. **b** Focal spots reconstructed from serial wavefront modulation on the two planes within the time period of 1 μs. Contrast of focal spots in these images was improved due to the nine-fold averaging by physically translating the scattering medium and by using the previously developed digital background cancellation method[51]. Note that the actual instantaneous focal contrast is as shown in the left panel of Fig. 4b. Scale bar: 5 μm. **c** Simplified experimental schematic for large-volume and frequency-modulated focus control with two single-mode optical fibers (SMF1 and SMF2). Focal spots coupled into optical fibers are detected by photomultiplier tubes (PMT1 and PMT2) **d** Optical signals for focal spots simultaneously modulated at different frequencies that were measured through SMF1 and SMF2 placed at $(x, y, z) = (0, 2.5, 2)$ mm and $(0, 0, 2)$ mm, respectively. **e** Frequency spectrum of the optical signals in (**d**).

achieves random-access control of micrometer-sized focal spots over a large addressable volume of $5\text{ mm} \times 5\text{ mm} \times 5.5\text{ mm}$ at a maximum refresh rate of 31 MHz. The demonstrated results represent an improvement of more than one order of magnitude over existing varifocal lens schemes in terms of the refresh rate and tuning power.

However, unlike conventional lenses, the scattering-assisted lens does not directly form the image of the entire 2D plane at a desired depth, preventing their use in single-shot imaging applications such as bright-field microscopy. In addition, in contrast to conventional varifocal lenses, the scattering-assisted holographic focusing is typically associated with background intensity fluctuations and lower transmittance. Because the FLASH technique increases the modulation speed at the cost of DOF$_{spatial}$, it intrinsically yields a small focal contrast (i.e., the peak-to-background ratio), which would be a limiting factor in imaging applications (see Supplementary Text 4 for the implication of the small focal contrast). More specifically, the aggregated background signals from the speckle granules around a focal spot deteriorate the image contrast (see Supplementary Text 5 and Supplementary Fig. 12 for the demonstration of scanning fluorescence imaging based on the FLASH technique).

Nevertheless, the holographic focusing scheme provides distinctive advantages over conventional varifocal lenses, including random-access focusing, high tuning power, and a large transversal FOV[24]. Considering that the FLASH focusing technique can selectively scan target positions without wasting time on continuous scanning (e.g., raster scanning), it would be well suited for non-imaging applications in which the regions (or spots) of interest are sparsely distributed in a large volume, such as in laser micromachining[3,22], optical tweezers[43], and cell-targeted activity monitoring and stimulation processes[44]. In particular, when nonlinear characteristics of materials and biological molecules are involved such as in optogenetic stimulation[45] and tissue ablation[46], a specific region in a large volume can be addressed at time with high spatial selectivity even with the existence of background speckle granules (see Supplementary Text 4 for the detailed description on the potential applications of the FLASH technique).

The FLASH technique, as a versatile holographic display, is broadly applicable to various light manipulation techniques, such as for complex point spread function engineering[47], for non-mechanical wide-angle beam steering for LiDAR[48]. For instance, with the FLASH technique, interferometric beam steering methods can simultaneously provide wide angular range and ultrafast steering speed, albeit with the reduced beam throughput (e.g., the energy efficiency of $10^{-6}$ in the previous demonstration based on a disorder-engineered metasurface[48]). The demonstrated scheme of binarized TM measurements also paves the way toward achieving real-time closed-loop wavefront shaping through a highly dynamic scattering medium, only requiring 100 μs to characterize a TM with ~1000 input modes. For comparison, state-of-the-art wavefront shaping systems require more than 1 ms for TM measurements of the same size[49,50].

The energy efficiency of the FLASH focusing technique was measured to be on the order of $10^{-5}$. This low energy efficiency can be easily improved by one order of magnitude by using a wider-FOV objective lens and 4K-resolution DMD (i.e., by using larger $N$). Moreover, the use of a pulsed laser can significantly reduce the energy loss and the undesired interference effect within the transient time of a line beam traveling across neighboring columns. Additionally, the spatial characteristics and energy efficiency could be greatly improved using an engineered scattering medium instead of the opal diffuser glass used in this work[24]. In particular, using metasurface technology, the scattering profile can be precisely controlled to achieve optimal transformation for specific manipulation tasks while also realizing the benefits of high transmittance and long-term stability. The focus control speed can be enhanced as well by incorporating a telecentric F-theta lens for line beam projections and a faster 1D scanner such as a polygon scanner. Assuming the scan rate of a typical 24-facets polygon mirror, $f_{scan} = 60$ kHz, and the number of columns $M$ of 2,000 with the extended FOV, the proposed scheme can practically achieve a modulation speed of 120 MHz (see Supplementary Text 1 for the detailed description on the optimal implementation of the FLASH technique).

To conclude, the present work has demonstrated that the conventional speed limit in spatial light modulator technology can be bypassed by exchanging the spatial DOF for the gain in the temporal DOF. We anticipate that the ability to handle DOFs in two domains in an interchangeable and tunable manner will pave the way for novel opportunities to investigate or manipulate systems that exhibit high spatial and temporal (or equivalently, spectral) complexity.

## Methods

### Detailed experimental setup

The overall experimental setup is depicted in Supplementary Fig. 2a. We used a 532 nm green laser (Verdi G5 SLM, Coherent) as the laser source. A collimated laser beam was split into two laser beams with a beam splitter. A laser beam traveling downward served as the reference beam for measuring the TM of a scattering medium using a phase-shifting holography technique. The reference beam's phase was controlled by a DMD (V-7001, Vialux). Meanwhile, a laser beam traveling to the right was employed for wavefront shaping. This beam was initially converted into a line beam using a cylindrical lens (ACY254-100-A, Thorlabs) and then relayed onto another DMD (V-7001, Vialux) through an achromatic lens and a 4x objective lens (PLN4X, Olympus). The RS used here was positioned on the back focal plane of the objective lens to undertake the scan of the line beam on the DMD. We introduced a volume holographic grating (WP-360/550-25.4, Wasatch Photonics) between the objective lens and the DMD to establish a retroreflective configuration. where the tilt angle of each DMD micromirror was offset by the first-order diffraction angle of the line beam. It is worth noting that the diffraction efficiency of the first-order beam diffracted through the holographic grating was measured and found to be ~75% at perpendicular incidence. After binary modulation of illuminated columns of the DMD, the modulated line beam propagated backward along the incident optical path and was expanded back into the collimated beam through the cylindrical lens and RS, ensuring a fixed projected area regardless of the line beam position on the DMD. Here, the measured light utilization efficiency of our modulation method, defined as the ratio of the input power to the output power, was found to be around 10%. The input power was measured on the left side of the polarized beam splitter, while the output power was measured on the projection plane at the scattering medium's surface. Spatial filtering took place between two achromatic lenses (L2 and L3) to remove unwanted diffracted beams from the DMD. The holographically focused beam behind the scattering medium with wavefront shaping was imaged on a CMOS camera (acA1440-220um, Basler) using a microscopic setup consisting of a 60x objective lens (MPLA-PON60X, Olympus) and a tube lens (TTL200, Thorlabs). For the demonstration of MHz modulation speed in Figs. 3e, 4c, a pinhole with a diameter of 2 μm was positioned at the target plane of 2 mm behind the scattering medium and the transmitted beam through the pinhole was detected by the PMT (PMT1001, Thorlabs) placed in the conjugate plane of the camera. The analog voltage signal from the PMT was recorded by an oscilloscope (Tektronix, TBS2102B). The all-voltage signals shown in this work were low-pass filtered with a cutoff frequency of 40 MHz to eliminate spike-like noise signals due to stray light.

### Alignment of the line beam on the DMD

We detail the process for implementing the precise alignment of the line beam onto a specific column of the DMD. First, we constructed an alignment unit comprising a pellicle beam splitter, a 4× objective lens, and a camera within the experimental setup to visually monitor the amplitude-modulated line beam generated by the DMD. Using this alignment unit, we performed spatial mapping between the DMD plane and the camera plane. During this process, the cylindrical lens was temporarily removed from the setup, allowing a collimated beam to illuminate the entire region of the DMD. The mapping between the

two planes was achieved by displaying a specific binary pattern on the DMD and physically adjusting the spatial position and tip/tilt of the camera device using goniometric and translational stages. Upon completing the mapping, the cylindrical lens was reinserted into the setup to illuminate the line beam on the DMD. To ensure precise alignment of the line beam, the cylindrical lens was physically rotated with a rotation stage until the line beam was accurately aligned parallel to the center column on the DMD. By operating the RS, we confirmed that the scanned line beam could be parallel to every DMD column. Precise implementation of this alignment procedure is essential for accurately achieving 1D spatial modulation from a single DMD column without unwanted 1D modulation from neighboring columns.

### Width of the line beam scanned on the DMD
To achieve 1D modulation by a single DMD column for $M_{col} = 1$ without unwanted modulation (i.e., crosstalk) from adjacent columns, the width of the line beam in the scanning direction must be narrower than the DMD pixel size. To verify that this condition is fulfilled, we monitored the intensity profile of the line beam on the DMD plane using a microscopic setup (Supplementary Fig. 3a, b). Supplementary Fig. 3c shows the FWHM for the 1D profile of the line beam in the scanning direction. The FWHMs of the line beam were confirmed to be narrower than each micromirror pitch of 13.7 µm across the entire scanning range of around ± 2.5 mm (i.e., the physical width of all active DMD columns). For $M_{col}$ values higher than 1, the width of the line beam illuminated on the DMD was properly adjusted by controlling the diameter of the incident beam onto the 4× objective lens in front of the DMD.

### Correction of column-wise wavefront distortion
We describe here the process of correcting column-dependent wavefront distortions. Each modulated beam from individual DMD columns propagates along different optical paths between the RS and the DMD. Moreover, there is wavefront curvature across the entire DMD surface. Due to these factors, the modulated beam experiences column-dependent wavefront distortion. In this experiment, we corrected these wavefront distortions using a typical iterative optimization method based on Zernike modes. Initially, we measured the TM of the scattering medium using the central $m$th column of the DMD without running the RS. We then calculated the phase pattern $\boldsymbol{\varphi}_m$ based on the measured TM and displayed its binary amplitude pattern $\mathbf{A}_m$ on the $(m + 1)$th column for focusing at a specific position behind the medium. Upon starting the RS, we would observe a focal spot with a slightly lower peak intensity compared to that for the $m$th column, as the $(m + 1)$th DMD column introduces a small amount of wavefront distortion. To correct this distortion and recover the degraded peak intensity, we optimized the correction pattern $\mathbf{C}_{m+1}$ using several Zernike modes to maximize the spot intensity at a specific position on the camera or on the photodetector. We only used first-order and second-order Zernike modes representing 'defocus' and 'astigmatism' aberrations along the major axis of the line beam (i.e., modulation direction). After correcting the distortion on the $(m + 1)$th column, the amplitude pattern $\mathbf{A}_{m+1}$, converted from the superposition of the focusing pattern $\boldsymbol{\varphi}_m$ and the correction pattern $\mathbf{C}_{m+1}$, was displayed on the $(m + 2)$th column. Similarly, the correction pattern $\mathbf{C}_{m+2}$ on the $(m + 2)$th column was optimized using two aberration modes to offset the wavefront distortion. By repeating this type of optimization for every column, we eventually acquired a set of wavefront correction patterns $\{\mathbf{C}_1, \mathbf{C}_2, \mathbf{C}_3, \ldots, \mathbf{C}_M\}$ for all columns. In practice, as correction patterns between neighboring columns are highly correlated, it was not necessary to calibrate all columns individually. Instead, we measured wavefront correction patterns $\{\mathbf{C}_1, \mathbf{C}_{11}, \mathbf{C}_{21} \ldots, \mathbf{C}_M\}$ for every ten columns and calculated other correction patterns by means of interpolation.

### Continuous ultrafast spatial modulation
Here, we describe the process of synchronously updating DMD patterns in accordance with the movement of the RS for continuous spatial modulation. First, it is crucial to note that the DMD needs a finite time to refresh the frames. Specifically, since the DMD controller operates at a 400 MHz clock speed and takes 40 ns to load a single row with 16 clocks, loading a binary image consisting of all 352 rows in the active DMD area requires ~14 µs. After loading, each micromirror takes around 4 µs to change its state and an additional 8 µs to mechanically settle down. Considering all these time intervals, a complete frame refresh requires a total of about 26 µs. To allocate sufficient time for the continuous updating of DMD frames upon every directional change of the RS, we set the RS angular range to be ~2.7 times broader than the minimum angular range precisely matched to the active DMD area of $M = 340$. With this setting, we can safely reserve around 30 µs, which is longer than the DMD frame refresh time (26 µs).

Supplementary Fig. 10a, b show the system control diagram and the electrical signal flow diagram, respectively. The analog voltage output from channel 1 (CH1) of the function generator (Tektronix, AFG1062) controls the amplitude (i.e., angular range) of the RS. The synchronization signal with a frequency of 12 kHz from the RS controller is input to the trigger input terminal of the function generator (FG). Upon detecting the rising edge of the synchronization signal from the RS, CH2 of the FG outputs a 24 kHz pulse train to the trigger input terminal of the DMD. In this experiment, the time delay between the RS synchronization signal and the pulse train is adjusted to 30 µs, enabling the DMD to start refreshing as the RS passes through the DMD active area. Once the DMD detects the rising edge of the pulsed signal from the FG, it spends the first 26 µs on a complete refresh, which is followed by stable projection time for the next 13 µs. CH2 of the FG is also connected to the trigger input terminal of the oscilloscope (Tektronix, TBS2102B) to control the timing of the acquisition of the analog voltage signal from the PMT. The output of the PMT is connected to CH1 of the oscilloscope to record the voltage signal waveform.

Lastly, we consider the available number of 1D patterns for continuous modulation under the current setting. A DMD module can only achieve an update rate of around 24 kHz when using binary frames stored in on-board memory. Thus, the total number of available 1D patterns can be calculated as the product of the number of prestored binary frames and the number of active columns on each frame. In the current setting, as a total of 175,000 binary frames can be prestored with 16 GB of on-board memory, each containing 340 columns, the total number of available 1D patterns is around 60 million. This total number could be further improved by streaming new frames from the computer into the on-board memory while sequentially displaying the prestored frames on the DMD from memory.

## Data availability
The data that support the plots and other findings in this study are available from the corresponding authors upon request.

## Code availability
The MATLAB codes used in this work are available from the corresponding authors upon request.

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

## Acknowledgements

This work was supported by JST-FOREST (Grant No. JPMJFR205E) to A.S. and JST-CREST (Grant No. JPMJCR1656) to Y.S., by JSPS KAKENHI (Grant No. JP21H00404 and JP21H02446 to Y.S. and JP21H01393 to A.S.), by the Nakajima Foundation to A.S., the Research Foundation of Opto-Science and Technology to A.S., by Konica Minolta Science and Technology Foundation to A.S., by the Ozawa and Yoshikawa Memorial Electronics Research Foundation to A.S., by the Cooperative Research Project of Research Center for Biomedical Engineering to A.S., by the National Research Foundation of Korea (NRF) grants funded by the Korea government (MSIT) (Grant No. NRF-2021R1A5A1032937 to G.S. and M.J. and 2021R1C1C1011307 to G.S. and M.J.), by the Samsung Research Funding and Incubation Center of Samsung Electronics grant SRFC-IT2002-03 to G.S. and M.J., by the LG Electronics-KAIST Digital Healthcare Research Center to M.J., by the Korea Agency for Infrastructure Technology Advancement (KAIA) grant funded by the Ministry of Land, Infrastructure and Transport (grant RS-2021-KA163379) to M.J., by the Photo-excitonix Project in Hokkaido University to H.M., and by Crossover Alliance to Create the Future with People, Intelligence and Materials from MEXT to A.S. and H.M. We are grateful to the Nikon Imaging Center at Hokkaido University for the preparation of biological specimens in scanning fluorescence imaging.

## Author contributions

A.S. and M.J. conceived of the initial idea. A.S. and M.J. expanded and developed the concept. A.S. and M.J. developed the theoretical model and designed the experiments. A.S. carried out all experimental processes and analyzed the experimental data with the help of R.H. and M.J. A.S. and M.J. wrote the manuscript with the help of R.H., G.S., H.M., and Y.S. H.M., Y.S., and M.J. supervised the project.

## Competing interests

The authors declare no competing interests.
