## [Peer Review File · Nature Communications]

Large-volume focus control at 10 MHz refresh rate via fast line-scanning amplitude-encoded scattering-assisted holographyREVIEWER COMMENTS

Reviewer #1 (Remarks to the Author):

In the work "Large-volume focus control at 10 MHz refresh rate via fast line-scanning amplitude-encoded scattering-assisted holography", Shibukawa et. al. present a method, termed FLASH, that trades off the high spatial degree-of-freedom (DOF) of a spatial light modulator for an improved temporal DOF. By utilizing a random scattering medium and line scanning over a DMD, the focus of the optical system can be addressed to the targeted position in a 3D volume. This wavefront engineering based scanning paradigm, as the authors suggested, achieves an ultra-high modulation speed as it alleviates the inertia problem in those mechanical scanning based method.

I think this is an interesting work as it offers great flexibility over the scanning speed and the quality of the focused beam. The principle of the proposed method is clearly explained in the manuscript. Nonetheless, I have some concerns over the practical applications of this method that I hope the authors could address in their revision.

Major points:

1. As the authors explained, the focus quality of FLASH depends on the number of free modes used in its generation process. In their experiment, the active area of the DMD device used for wavefront shaping was 340x352. I wonder why the authors chose to use such a small number of pixels. Is the magnification of the lens the main problem? If that's the main problem, I wonder why the authors didn't try to demonstrate it with a suitable lens as it seems to be relatively easy fix. Will a f-theta lens be a more appropriate choice over objectives for FLASH?

I also noticed that the model of the DMD in the wavefront shaping arm is missing in the manuscript. The authors might have used the same type of DMD for both reference arm and the wavefront shaping arm. But I think it needs to be clarified in the manuscript.

2. As the number of free modes used for generating a specific focus is low, the peak-to-background ratio (η in the paper) of the formed focus is also small. This might be a problem in real applications as the aggregated background signal might be well above the desired signal from the focus. This is also one of my biggest concerns with respect to the practical application of FLASH. I wonder how a scanned image in FLASH would look like, especially for a non-binary object. With this, the readers can have a rough idea of how the final image will look like if they implement FLASH in real life.

One following question is that the behavior of the focus seems to be irregular and unpredictable (Figs. 3e, 4c, 5d). The maximal signal of the focus fluctuates quite a bit. I wonder if the author knows the reason of the fluctuation. Another thing I noticed is that the signal at the "random" state is much higher than that calculated from η . It is also valuable to discuss why we see a higher signal in those cases.

3. In measuring the transmission matrix, was the camera placed at a fixed axial position in the entire measurement? It seems to me that if 3D addressable focus is desired, transmission matrices at different axial position are useful for optimizing the focus quality. So, I wonder if FLASH measures transmission matrices at different axial position or it measures at one position and relies on memory effect to scan the focus across a large axial range. If memory effect is applied, what is the range for the scattering medium used in the experiment?

4. It might be fruitful to discuss on using a pulsed laser as the light source. As the authors suggest, one can use multiple lines for wavefront shaping to obtain a high-quality focus. For a CW laser, the power is simply wasted in the transiting phase as the signal in the transition phase would be a mix of two independent foci. This would be an issue since the efficiency of FLASH is considerably low. Synchronizing the pulsed laser with the scanning, one can minimize the loss.

5. I find supplementary figure 3b a bit confusing. I am not sure if the position in scan direction

refers to the displacement of the generated focus. In principle, the focus should be independently addressable and there is no correspondence between the direction of the scan and the direction of the focus' shift. When I saw the labels sitting above the figure, I thought this was the actual dimension of this figure. However, this interpretation is wrong as the scale bar is way smaller than that. Clarification is needed for this figure.

6. I sometime misread the inset frequency response figure (such as Fig. 4c) as the zoomed version of the time sequence. I think choosing a different line style (or simply changing the color) may help for a clearer presentation.

7. It might be better to write the factor for binary phase modulation as $1/(2\pi)$ or $(2\pi)^{-1}$ to avoid ambiguity. This applies to other numbers in the manuscript. But I think this can be corrected in the copyediting.

Overall, I think this is an interesting work for high-speed, random-access focus. The flexibility tuning between high refreshing rate and the quality of the focus. It can be a good complement to existing random-access techniques once my concerns are addressed.

Reviewer #2 (Remarks to the Author):

In this manuscript, the authors detail a scheme ("FLASH") for fast focus shifting and deflection at ~ 10 MHz rates, approximately one order of magnitude faster than currently reported state of the art (e.g. galvanometer scanners). The authors then provide a very robust demonstration of their scheme demonstrating dynamic on-demand focal shifting at millimeter scale volumes. The experimental documentation both in the manuscript and supplementary materials is commendably thorough. While the idea of spatially/temporally multiplexing a DMD/SLM with a cylindrical lens and actuator of some form is not particularly new (e.g. [1,2]), the authors achieve a commendably high sampling rate and then combine this with a scattered media focusing element for dynamic focal scanning in a volume. The combination of spatiotemporal multiplexed SLM with a scattering medium as a holographic focusing element at high speeds is novel and interesting, and I would recommend publication.

I do, however, have one comment for the authors pertaining to the discussion section. The claim was made that FLASH is suitable for "non-mechanical wide-angle beam steering for LiDAR", but to me this claim seems problematic. Generally, the performance of beam-steering systems can be characterized by the steering angular range, angular resolution, and number of resolvable beam directions, beam throughput, and speed [3-5]. Angular range, angular resolution, and number of resolvable directions are, amongst other things, a function of the number of elements within the beam steering system and the spacing of those elements [3-5]. The approach taken by the authors with FLASH is to take an $M \times N$ DMD and sample its rows one by one each duty cycle. This achieves a large gain in sampling rate, but also reducing the instantaneous number of elements available for beam steering to N . N elements is less than $M \times N$ elements of the standard DMD, and so there will be a degradation in some of angular range, angular resolution, and number of resolvable directions. I think that if the authors want to include "wide angle beam steering" as a potential application, including a discussion on the trade-offs of FLASH with respect to the standard DMD in terms of angular range and resolution would be warranted.

[1] Teng, D., Liu, L., Zhang, Y., Pang, Z., Chang, S., Zhang, J. and Wang, B., 2014. Spatiotemporal multiplexing for holographic display with multiple planar aligned spatial-light-modulators. *Optics Express*, 22(13), pp.15791-15803.

[2] Takaki, Y. and Okada, N., 2009. Hologram generation by horizontal scanning of a high-speed spatial light modulator. *Applied Optics*, 48(17), pp.3255-3260.

[3] Xu, J., Cua, M., Zhou, E.H., Horie, Y., Faraon, A. and Yang, C., 2018. Wide-angular-range and high-resolution beam steering by a metasurface-coupled phased array. *Optics Letters*, 43(21), pp.5255-5258.

[4] Guo, Y., Guo, Y., Li, C., Zhang, H., Zhou, X. and Zhang, L., 2021. Integrated optical phased arrays for beam forming and steering. *Applied Sciences*, 11(9), p.4017.

[5] Heck, M.J., 2017. Highly integrated optical phased arrays: photonic integrated circuits for optical beam shaping and beam steering. *Nanophotonics*, 6(1), pp.93-107.

Reviewer #3 (Remarks to the Author):

In this manuscript the authors propose and demonstrate a method to quickly generate a focus at a desired position. The basic idea is that light passing through a scattering medium can be brought into a focus if the incident wavefront has been suitably shaped (ref 29). If we want a focus at a different position we need a different incident pattern (in this case the authors used binary amplitude modulation, which has been shown to work surprisingly well in ref 35), so creating a focus at a desired position on demand can be reduced to select one of the pre-obtained patterns. The authors achieve this by having each pattern as a single column on a DMD, and then scanning to select the desired one.

I don't have much of a problem with the method proposed or with the experiments presented. They do a satisfactory job at supporting the claims. My only real problem with the paper is the absolute lack of any discussion on the "wavefront shaping" part of the experiment. Light is assumed to form a focus, and the details and limitations of this approach are never discussed. For instance I didn't manage to find any information about the scattering medium itself. What is its mean free path? How stable is it over time? Also the fact that this approach will intrinsically yield small "enhancement factors" (i.e. ever focus will always be surrounded by a visible speckle pattern) is just mentioned in passing.

In conclusion, this is an overall nice paper, but I really wish the authors discussed more in depth the wavefront shaping part of the experiment.

Reviewer #1 (Remarks to the Author):

In the work “Large-volume focus control at 10 MHz refresh rate via fast line-scanning amplitude-encoded scattering-assisted holography”, Shibukawa et. al. present a method, termed FLASH, that trades off the high spatial degree-of-freedom (DOF) of a spatial light modulator for an improved temporal DOF. By utilizing a random scattering medium and line scanning over a DMD, the focus of the optical system can be addressed to the targeted position in a 3D volume. This wavefront engineering based scanning paradigm, as the authors suggested, achieves an ultra-high modulation speed as it alleviates the inertia problem in those mechanical scanning based method.

I think this is an interesting work as it offers great flexibility over the scanning speed and the quality of the focused beam. The principle of the proposed method is clearly explained in the manuscript. Nonetheless, I have some concerns over the practical applications of this method that I hope the authors could address in their revision.

We thank the Reviewer for the concise description of our work along with many positive remarks. In the light of the Reviewer's comments on the practical applications (including **R1C3** below), we have conducted additional experiments on scanning fluorescence imaging and examined the usability of the proposed method in imaging applications. In particular, we have carefully discussed the limitations in imaging applications caused by the low focal contrast (i.e., peak-to-background ratio, PBR) and provided potential strategies to resolve them. In addition, we have provided specific examples in non-imaging application areas, which have been explored in previous works based on an interferometric focusing scheme, where high PBR is not necessarily required. We hope these efforts clarify the practical application scenarios of the proposed technique.

R1C1

As the authors explained, the focus quality of FLASH depends on the number of free modes used in its generation process. In their experiment, the active area of the DMD device used for wavefront shaping was 340x352. I wonder why the authors chose to use such a small number of pixels. Is the magnification of the lens the main problem? If that's the main problem, I wonder why the authors didn't try to demonstrate it with a suitable lens as it seems to be relatively easy fix. Will a f-theta lens be a more appropriate choice over objectives for FLASH?

We deeply appreciate the Reviewer's suggestion. The active number of DMD pixels is one of the critical factors that determines the overall performance of the proposed technique. The number of active columns, M , and rows, N , determines the gain in refresh rate and the PBR, respectively. Here, we describe the rationale behind our experimental setting and discuss various ways to implement an optimal line beam projection system that is capable of scanning a line beam as thin as a single DMD column across the entire DMD surface.

Our Experimental Setting

In our experiments, we used a standard objective lens (Olympus PLN4X, magnification 4, field number 22, numerical aperture 0.1) to project a scanning line beam onto the DMD plane. This lens has a field of view of 5.5 mm, which corresponds to ~400 columns or rows of the DMD used (Vialux V-7001, pixel size 13.7 μm). As suggested by the Reviewer, the high magnification

of the objective lens results in the limitation of the number of active columns or rows of the DMD. We chose this setting to minimize the width of the line beam by tightly focusing an incident beam, and thereby, to minimize the transient time for a line beam travelling across neighboring columns (i.e., rise and fall time of focus modulation). Additionally, the objective lens used is preferable because its pupil plane is positioned well outside the lens barrel so that we could directly position the 1-D scanner at the objective pupil plane without a pair of a scan lens and a tube lens and thereby could avoid the effect of aberrations due to these additional lenses.

Optimal Experimental Setting

The DMD used has 1024×768 pixels with a pixel size of $13.7 \mu\text{m}$. Therefore, it is desirable for the objective lens to have a scanning field of view (FOV) of around 14 mm and a spatial resolution of less than $13.7 \mu\text{m}$ to use the entire DMD surface for FLASH focusing with the setting of $M_{col} = 1$ (i.e., the number of columns illuminated with a single line beam is 1). There are commercially available 1X objective lenses that meet those requirements, such as M Plan Apo 1X from Mitutoyo and 1X ICO from Navitar.

Also, as the Reviewer suggested, the F-theta lens could be a good candidate because it can meet above requirements as well. As the proposed technique is based on the retro-reflecting configuration, a conventional non-telecentric f-theta lens would not serve the purpose. For example, the one from Ronar Smith (TSL-532-15-58) would be a telecentric F-theta lens that meets the requirement. Given that the f-theta lens is telecentric-type and the angular scanning speed is constant as in polygon scanning mirrors, it provides the additional benefit of a constant refresh rate of focal spot because the scanning speed of the line beam over DMD columns becomes proportional to the angular scanning speed.

Because of the trade-off relation between FOV and numerical aperture that apply to standard objective lenses, we expect that it becomes more challenging to simultaneously achieve a short transient time (i.e., line beam that is sufficiently thinner than the width DMD column) and a larger FOV with the above-mentioned alternatives. In this case, as insightfully suggested by the Reviewer in the comment R1C6, one may synchronize a pulsed source with a 1D scanner to mitigate the undesired effect (i.e., laser power loss) of long transient time.

As it requires significant modifications in the optical setup and resource-demanding troubleshooting processes, especially regarding wavefront correction, to figure out the best working configuration under the interplay between the transient time, the FOV, and the degree of wavefront distortion, we currently plan for a separate follow-up study to further optimize the performance of the FLASH technique.

We admit that there is a room to improve the PBR and the refresh rate by the factor of two to three with a better implementation of a line beam projection system. Such improvements would indeed be critical to achieve optimal performance for specific applications. Therefore, we have added Supplementary Text 1 presenting the detailed discussion on an optimal line beam projection system as a guidance in configuring a FLASH focusing system (Item 1 of List of Changes).

R1C2

I also noticed that the model of the DMD in the wavefront shaping arm is missing in the manuscript. The authors might have used the same type of DMD for both reference arm and the wavefront shaping arm. But I think it needs to be clarified in the manuscript.

We thank the Reviewer for pointing out this important omission. Indeed, we used the same type of the DMD (V-7001, Vialux) for the both reference and wavefront shaping arms. We have added the model information in the revised manuscript (Item 2 of List of Changes).

R1C3

As the number of free modes used for generating a specific focus is low, the peak-to-background ratio (η in the paper) of the formed focus is also small. This might be a problem in real applications as the aggregated background signal might be well above the desired signal from the focus. This is also one of my biggest concerns with respect to the practical application of FLASH. I wonder how a scanned image in FLASH would look like, especially for a non-binary object. With this, the readers can have a rough idea of how the final image will look like if they implement FLASH in real life.

We appreciate the Reviewer's insightful comments and suggestions. Indeed, the limited peak-to-background ratio (PBR) would be the most critical point when considering the practical applicability of the proposed technique. In the light of the comment from the Reviewer, we have conducted scanning fluorescence imaging with the FLASH focusing technique and examined the effect of aggregated background in imaging applications.

The non-zero background is the general feature that is commonly observed in wavefront shaping techniques for scattering media (i.e., interferometric focusing) due to the non-unitarity of associated transmission matrices. The usability of interferometric focusing schemes has been explored extensively since early pioneering works on complex wavefront shaping and optical transmission matrix. Here, based on the demonstrations from those studies, we discuss potential usability of the FLASH technique in two aspects – (i) non-imaging and (ii) imaging applications. Then, we articulate (iii) the results from our additional experimental demonstration on scanning fluorescence imaging.

(i) Non-imaging Applications

At the current stage, we consider that the FLASH focusing technique would be best suited for non-imaging application areas in which some nonlinear processes that can reduce the effect of speckle backgrounds are involved and a specific region within a large volume is important at a time (i.e., not all the positions of the specimen are simultaneously addressed). There are many non-imaging application areas with such characteristics, including laser-scanning micromachining, optical tweezers, and photoactivation of biological activities (e.g., optogenetics, ablation, and so on).

As described above, scattering-aided interferometric focusing schemes including the FLASH technique accompany with surrounding background fluctuations, but they have proven to be practically useful in many previous studies. For instance, deep tissue optogenetic stimulation of a targeted single neuron has been demonstrated with the PBR of ~ 2 [*H. Ruan et al., "Deep tissue optical focusing and optogenetic modulation with time-reversed ultrasonically*

encoded light”, *Science Advances* 3, eaao5220 (2017)]. In this study, although the PBR was extremely low (approximately 2), the interferometric focusing could be practically used based on the nonlinear characteristic of optogenetic proteins (i.e., switching behavior over a threshold intensity). Similarly, optical trapping through a scattering medium [T Cizmar, M Mazilu, K Dholakia, “In situ wavefront correction and its application to micromanipulation”, *Nature Photonics* 4, 388 (2010)] and tissue ablation [E Kakkava et al., “Selective femtosecond laser ablation via two-photon fluorescence imaging through a multimode fiber” *Biomedical Optics Express* 10, 423 (2019)] have also been successfully demonstrated.

(ii) Imaging Applications

The FLASH focusing technique, like other scattering-aided interferometric focusing methods, accompanies background intensity fluctuations, which we consider is the key disadvantage in comparison to conventional varifocal lenses. Especially, in our proposed scheme, the PBR is further compromised to boost the speed of modulation, resulting in the experimental PBR of around 40 at the setting of $M_{col} = 1$. In that regard, we fully agree with the Reviewer’s concern that the signal from the focal spot position will be obscured by the aggregated signal from the surrounding background regions.

We think that the practical use of the FLASH focusing technique for imaging applications (or, in a broad sense, the use of interferometric focusing for imaging applications) is the subject of future research. Only a few research efforts have been made to use the interferometric focusing scheme for imaging so there is no standardized imaging scheme to test our proposed scheme. Considering imaging systems is in large comprised of an illumination path and detection path, the interferometric focusing scheme only serves the illumination path and the optimal configuration for the detection path has not been fully explored yet. Here, we first present three potential ways to implement detection paths for scanning imaging systems with the FLASH focusing technique.

1) Confocal or quasi-confocal configuration in transmission mode: This strategy is a straightforward approach. One may configure the optical sensors in confocal or quasi-confocal positions to suppress the signal from background regions [IM Vellekoop, CM Aegerter, “Scattered light fluorescence microscopy: imaging through turbid layers”, *Optics Letters* 35, 1245 (2010) and M Jang et al., “Wavefront shaping with disorder-engineered metasurfaces”, *Nature Photonics* 12, 84 (2018)]. We believe that this scheme is most relevant to the context of the Reviewer’s concern and suggestion to provide “*how a scanned image in FLASH would look like*” in the presence of fluctuating backgrounds. In this strategy, the image contrast is highly dependent on the PBR and the size of the collection area, which is discussed in the following section with the additional experimental results.

2) Wavefront shaping for epi-detection scheme: Because of scattering media, conventional epi-detection schemes (i.e., scan and descanned with same scanning mirrors) cannot be directly applied in interferometric focusing schemes. However, as demonstrated in [IM Vellekoop, M Cui, C Yang, “Digital optical phase conjugation of fluorescence in turbid tissue”, *Appl. Phys. Lett.* 101, 081108 (2012)], a detection path can also be implemented with wavefront shaping techniques. In this study, it has been shown that the narrow-band signal of fluorescence emission from a targeted position can be selectively detected via the phase

conjugation process. This configuration would provide the confocal image with the contrast of the PBR for excitation multiplied by the PBR for detection.

3) Multiphoton excitation: We anticipate that multiphoton fluorescence imaging is one of the promising ways to implement an imaging system that circumvents the low PBR issue. The PBR of the fluorescence signal emitted in the two-photon excitation process would be at least (neglecting the effect of temporal focusing) larger than the square of the PBR of the focal spot made with interferometric focusing schemes. Like a conventional two-photon microscope, we expect that one may retrieve a high-contrast image with a photomultiplier tube that collects a total amount of fluorescence as demonstrated in [U Weiss, O Kata, “Two-photon lensless micro-endoscopy with in-situ wavefront correction”, *Optics Express* 26 28808 (2018)]. We note that, in this study, the PBR of the focal spot was around 300 (i.e., the effective PBR of the fluorescence signal was $\sim 10^5$) which is achievable by adjusting M_{col} in our setup or using a commercial DMD with a high resolution of 1920×1080 .

(iii) Scanning Fluorescence Imaging with FLASH focusing

As described above in “1) Confocal or quasi-confocal configuration in transmission mode”, we performed scanning imaging of fluorescent microspheres and fluorescence-stained biological specimens with the FLASH focusing technique to provide “*how a scanned image in FLASH would look like*” and also to show that the practical use of the FLASH based on this imaging scheme is potentially useful in real life.

1) Experimental setting and image acquisition process

As illustrated in **Figs. R1a** and **R1b**, the fluorescence excited at each scan position over a raster-scan path was captured by the camera in a microscope setup and the fluorescence intensities integrated over a collection area of a certain diameter φ_{pin} (green circle in **Fig. R1b**) in the captured image was taken as the value of the corresponding pixel in the scanning image (green square in **Fig. R1b**). After repeating this procedure for all scan positions along the entire raster-scan path, the scanning fluorescence image was constructed. This imaging scheme is similar to that of conventional confocal microscopy, except that background signals from out-of-focus positions are digitally removed by setting up a pinhole-like collection area in the captured image of the camera.

2) Varying collection area and its practical implication for imaging application

We have demonstrated the capabilities of the MHz scanning speed and the addressability to millimeter-scale volume using the FLASH technique. One potential configuration in the detection path that can take advantage of those capabilities in imaging application would be to use a low-magnification objective lens and a high-speed camera with a MHz frame rate. As the practical example, our previous study based on the scattering-assisted focusing scheme used a standard 4x objective lens (whose NA is 0.1) with a field number of 24 to demonstrate scanning fluorescence imaging with the effective NA of 0.5 over a wide field-of-view (FOV) of 6 mm diameter that is inaccessible in conventional

microscope [M Jang et al., “Wavefront shaping with disorder-engineered metasurfaces”, *Nature Photonics* 12, 84 (2018)]. Instead of obtaining a beneficial wide-FOV, the use of the low magnification objective in the detection path results in the spatial resolution of the detection system (i.e., the detection PSF) that is sufficiently worse than that of the illuminated focal spot (i.e., the illumination PSF), meaning that the integration of fluorescence intensities over the collection area whose diameter is set to the size of the detection PSF includes not only the signal from the focal spot but also the noises from surrounding aggregated backgrounds.

Figure R1 | Scanning fluorescence imaging with FLASH focusing. **a**, Optical setup. **b**, Schematic of the method to construct scanning fluorescence image. **c-f**, Scanning fluorescence images of fluorescent microspheres with different collection diameters. Scalebar: $1 \mu\text{m}$. **g**, Wide-field fluorescence image of the fluorescent microspheres. Scalebar: $1 \mu\text{m}$. **h-k**, Scanning fluorescence images of fluorescence-stained HeLa cells with different collection diameters. Scalebar: $1 \mu\text{m}$. **l**, Wide-field fluorescence image of the HeLa cells. Scalebar: $1 \mu\text{m}$.

In the following scanning fluorescence imaging experiment, we varied the diameter of the collection area (i.e., varying the diameter of the pinhole in a quasi-confocal configuration) to see the effect of aggregated backgrounds in a scanning image and also to find out what magnification of objective lens can be chosen under the present PBR of the FLASH technique. Specifically, the diameter of the collection area, ϕ_{pin} , was set to 0.36 μm , 0.65 μm , 1.08 μm , 2.49 μm which corresponds to the FWHM of the diffraction-limited spot of off-the-shelf standard objective lenses with magnifications of 60x, 20x, 10x, and 4x. These values are based on standard objective lenses provided by Olympus (MPlanApo N 60x, UPlanFL N 20x, UPlanFL N 10x, UPlanFL N 4x).

3) Experimental results

For imaging fluorescent microspheres, with a spot size of 0.57 μm and a step size of 0.09 μm , the raster-scanning was performed over $11 \times 11 \mu\text{m}^2$, resulting in 120×120 illumination spots (i.e., image resolution). For the biological specimens, the raster-scanning was performed over $16 \times 16 \mu\text{m}^2$ with a step size of 0.18 μm , resulting in 90×90 image resolution. We used sparse sample and contiguous biological sample - diluted fluorescence microspheres (F13082, Invitrogen) and HeLa cells stained with a high-affinity F-actin probe conjugated to rhodamine fluorescence dye (ActinRed™ 555 ReadyProbes™ Reagent, Invitrogen).

The imaging results are shown in **Figs. R1c-I** along with the wide-field fluorescence images excited by a collimated laser beam illumination. In the tight confocal configuration of $\phi_{\text{pin}} = 0.36 \mu\text{m}$ corresponding to the FWHM of the 60x objective, the results present well-resolved images whose structural details are well-matched with the wide-field imaging results. In fact, as the confocal configuration provides depth-sectioning capability, the imaging results present finer structural details compared to the wide-field results. We note that, due to the low fluorescence level, the stochastic camera noise slightly degrades the imaging results for the biological sample. Even in the case of $\phi_{\text{pin}} = 1.1 \mu\text{m}$ corresponding to the FWHM of the 10x objective where the collection area is around 4 times ($= (1.08/0.57)^2$) bigger than the focal spot size, the image qualities are quite robust against the effect of background speckles. The PBR in our experimental configuration at $M_{\text{col}} = 1$ was around 40, which was consistent with our observation in imaging results. Those imaging results indicate that the combined use of the FLASH illumination and the detection system with the low magnification objective could result in a usable imaging contrast for the wide-FOV (e.g., 3 mm \times 3mm FOV for 10x objective) in the presence of the effect of the aggregated backgrounds. More specifically, under the current experimental configuration, the combined use of a low-NA objective lens and an ultra-high-speed camera with a MHz frame rate (e.g., HPV-X2, 10MHz frame rate at full resolution, Shimadzu) would be a possible detection system in the transmission geometry.

Lastly, as suggested by the Reviewer, when the collection area becomes significantly larger than the spot size, as shown in the case of $\phi_{\text{pin}} = 2.5 \mu\text{m}$ where the collection area is 20 times larger than the spot size, the aggregated backgrounds obscured the signal from the peak so that the structural details were missed. Compared to the sparse microsphere sample, this effect is more pronounced in the case of contiguous biological sample.

In the revised manuscript, we have further clarified the limit of low enhancement factor and the presence of fluctuating backgrounds in the Discussion section of the main text (Item 3 & 20 of List of Changes). We have also added the detailed discussion on practical applications, including non-imaging and imaging applications, of the FLASH focusing technique in relation to the effect of backgrounds as Supplementary Text 4 (Item 4 of List of Changes) with citations (Item 21 of List of Changes). Finally, we have presented our experimental procedures and results on scanning fluorescence imaging in Supplementary Text 5 and Supplementary Fig. 12 (Item 5 of List of Changes) to demonstrate the detrimental effect of the aggregated backgrounds in imaging applications in detail.

R1C4

One following question is that the behavior of the focus seems to be irregular and unpredictable (Figs. 3e, 4c, 5d). The maximal signal of the focus fluctuates quite a bit. I wonder if the author knows the reason of the fluctuation.

Another thing I noticed is that the signal at the “random” state is much higher than that calculated from η . It is also valuable to discuss why we see a higher signal in those cases.

Thank you for the careful examination of the plot. Based on a thorough consideration of experimental details, we attribute (i) the peak intensity fluctuation to the combined effect of - (1) intrinsic statistical fluctuation in the optical response of scattering media (i.e., transmission matrix), and (2) a large pinhole (or fiber core) size used to collect the peak intensity, both of which are associated with the degeneracy in binarized wavefront solutions for a single focal spot. (ii) The higher level at the “random” state in comparison to the “phase conjugation” state is also caused by (2) the large pinhole (or fiber) size used to collect the peak intensity. The detailed discussion is provided below.

(1) Intrinsic statistical fluctuation

The relation between input modes (i.e., DMD pixels) and output modes (i.e., independent speckle granules) can be described based on the transmission matrix formalism: $E_m^{\text{out}} = \sum_n t_{m,n} E_n^{\text{in}}$ where E_m^{out} and E_n^{in} are complex fields of the m th output mode and the n th input mode. $t_{m,n}$ is the transmission matrix element that relates the m th output mode with the n th input mode. The matrix elements, t_{mn} , are statistically independent and follow a complex Gaussian distribution.

The binary phase conjugation process used in our experiments is to selectively turn on the input modes that additively contribute to the intensity of a target output mode, $|E_{\text{target}}^{\text{out}}|^2$ (e.g., to choose input modes with $|\arg(t_{\text{target},n})| < \pi/2$). Therefore, roughly half of the input modes are set on and another half are set off. When “ensemble averaged”, this process results in the peak intensity, $\langle |E_{\text{target}}^{\text{out}}|^2 \rangle$, $1/2\pi \times N$ times enhanced compared to the background intensity, $\langle |E_{\text{background}}^{\text{out}}|^2 \rangle$. N is the total number of input modes.

In our experiments, we corrected wavefront distortion for different DMD columns to achieve an optimal peak intensity (as shown in Supplementary Fig. 7). Therefore, the binarized phase conjugation process is modified to the process of turning the input modes with $|\arg[t_{\text{target},n} \times \exp(i\Delta\varphi_{n,k})]| < \pi/2$ into the “ON”-state where $\Delta\varphi_{n,k}$ is the column-dependent

correction map (i.e., correction map for the k -th column). Therefore, although we optimized the wavefront for the same target focus, the 1D binarized patterns vary substantially for different DMD columns. This effect is evidenced in Supplementary Fig. 7 which shows the similar background patterns without wavefront correction (i.e., when the same 1-D binary pattern is repeated) and the varying background patterns with wavefront correction.

Due to this aspect, each realization of phase conjugation process for different DMD columns is subject to statistical fluctuation from the intrinsic stochastic property of the transmission matrix (i.e., statistical fluctuations in adding different sets of random phasors). The same fundamental stochastic noise should be observable when focal spots are created at different positions. Setting the peak intensity as $I_{\text{target}} = |E_{\text{target}}^{\text{out}}|^2$, the fluctuation level can be characterized as the ratio of the standard deviation, σ_I , to the average value, μ_I , of I_{target} . With the numerical simulation of repeated phase conjugation process through different transmission elements, we found that this fluctuation level is scaled with \sqrt{N} , which is consistent with the central limit theorem (see **Fig. R2**). The simulated fluctuation level for phase conjugation processes with phase-only and binary-amplitude modulations for different values of N is plotted with fitted curves in **Fig. R2**. For each value of N , phase conjugation is repeated 1,000 times to get the statistically valid fluctuation level. For our experimental case of binary-amplitude phase conjugation with $N = 352$, the fluctuation level is estimated to be 15 %.

Figure R2 | Simulated intensity fluctuation of a focal spot for different total number of input modes, N .

(2) Effect of large collection area

In the experiments for Figs. 3e, 4c, and 5d, the focal spot was created at the axial distance $z = 1.5 - 2.0$ mm. The full-width at half-maximum of the intensity profile was measured to be around 400 - 600 nm. To measure the temporal profile of the peak intensity, we placed a pinhole with a diameter of 2 μm directly at the target focal plane for Figs. 3e and 4c and a tip of a single-mode fiber with the acceptance NA of ~ 0.12 and the mode field diameter (MFD) of ~ 3.5 μm at the target focal plane for Fig. 5d. Then, we placed a photo-multiplier tube (PMT) roughly at the conjugate plane of the pinhole plane for Figs. 3e and 4c and at the conjugate plane of the distal end of the single-mode fiber for Fig. 5d.

(i) Regarding high signal level at random state (i.e., contrast reduction): In this experimental setting, the resultant intensity at the PMT was contributed from roughly 10 background optical modes around the peak (i.e., speckle granules), where the number of background modes M_{bcg} within a pinhole was roughly counted by the area ratio with the $1/e^2$ diameter of speckles. Therefore, the aggregated background signal from many optical modes contributed to the PMT signal during the “random” modulation period. In this case, the signal contrast between the “focusing” state and “random” state is decreased by ~ 10 times compared to the observed PBR from a camera shown in Figs. 2h and 4b. Considering that the experimental PBR at $M_{col} = 1, 2, 4$ were respectively around 40, 60, 140, the signal contrast would be estimated to be around 4, 6, 14 (i.e., the background level of 25 %, 17 %, 7 % compared to the averaged peak level). The actual signal contrasts in Fig. 3e and Fig. 4c are roughly consistent with our estimated contrasts. For Fig. 5d, the large collection area (i.e., the mode field diameter of the fiber larger than the size of the speckle granule) similarly degrades the signal contrast. Plus, the fact that two focal spots were simultaneously generated onto the two fiber tips additionally contributes to a higher background level. In this case, half of the superposed wavefronts contributes to each peak while the background level remains the same. Thus, the signal contrast would be further decreased by two times.

(ii) Regarding peak intensity fluctuation: If the measured intensity from the PMT is the intensity summation of peak intensity and backgrounds intensity, it can be described as $|E_{target}^{out}|^2 + \sum_{m \in M_{bcg}} |E_{bcg,m}^{out}|^2$ where $E_{bcg,m}^{out}$ is the background field of m th output mode and M_{bcg} is the set of optical modes within the pinhole area. In this case, the fluctuation level is mostly determined from the fundamental stochastic property of transmission matrix described above as the statistical fluctuation level of $\sum_{m \in M_{bcg}} |E_{bcg,m}^{out}|^2$ is much smaller than that of $|E_{target}^{out}|^2$.

However, the actual fluctuation level of the peak intensity shown in Fig. 3e is larger than the expected level (~ 15 %) of the above numerical simulation. We speculate that this discrepancy is attributed to the interferometric mixing of the complex field of the target focus with the background field in some degree. To put it in a mathematical term, the measured intensity can be expressed by $|E_{target}^{out} + \sum_{m \in M_{bcg(overlap)}} E_{bcg,m}^{out}|^2$. Here, depending on the relative phase between the target mode and the background modes, the fluctuational level

can be amplified because of the cross terms (i.e., interference terms) of $E_{\text{target}}^{\text{out}}$ and $\sum_{m \in M_{\text{bcg}}(\text{overlap})} E_{\text{bcg},m}^{\text{out}}$. For instance, as a simple back-of-envelope calculation, when the ensemble-averaged PBR is 100, the amplitude ratio of target mode and background mode is 10 so that, depending on their relative phase, the PBR can be fluctuated in between 81 (out-of-phase) to 121 (in-phase). While such interferometric mixing effect can be further mitigated by exactly aligning the PMT's active area in the conjugate plane of the pinhole plane for Figs. 3e and 4c, we cannot avoid it in coupling of high-frequency optical fields into a single-mode fiber with a low acceptance NA for Fig. 5d. We note that this effect is experimental noise caused by the difficulties in measuring the ns-scale temporal profiles of micrometer-sized focal spots. As evidenced in the additional experimental results on fluorescence imaging where the scanned image reflects the actual brightness of fluorescence targets (shown in **Fig. R1** in our response to **R1C3**), the actual spot intensity would not fluctuate in a such large degree.

Finally, as a minor factor, in Figs. 4c and 5d where M_{col} was set larger than 1, the line beam covers the full width of multiple columns so that even slight displacement results in a similar interferometric mixing effect of two output fields corresponding to two different foci (i.e. when line beam is positioned at the junction of two optimized regions on DMD). As suggested by the Reviewer in the comment R1C6, we anticipate that the integration of the FLASH technique with a pulsed light source would mitigate this effect.

In the revised manuscript, we have clarified the causes of the reduced signal contrasts and the fluctuation in peak intensity in the Results section (Item 6 of List of Changes) with the modified Methods section of "Detailed experimental setup" with additional experimental details (Item 7 of List of Changes). We note that the noises from a larger collection area are experimental ones while the fluctuation from the stochastic nature of scattering media represents a fundamental aspect of wavefront shaping technique. To describe this fundamental nature in full detail, we have added Supplementary Text 3 and Supplementary Fig. 9 (Item 8 of List of Changes).

R1C5

In measuring the transmission matrix, was the camera placed at a fixed axial position in the entire measurement? It seems to me that if 3D addressable focus is desired, transmission matrices at different axial position are useful for optimizing the focus quality. So, I wonder if FLASH measures transmission matrices at different axial position or it measures at one position and relies on memory effect to scan the focus across a large axial range. If memory effect is applied, what is the range for the scattering medium used in the experiment?

Thank you for raising this important point. In our experiment, to address focus at different axial positions, we repeatedly measured transmission matrices (TMs) with a camera placed at different axial positions (i.e., one TM at one axial position). We did not apply "memory effect of the scattering medium" because its angular range was practically too small. Indeed, we experimentally confirmed that the angular memory effect range of the opal glass is less than 0.1° (see Supplementary Fig. 6 for the experimental result on the memory effect range).

Although we have implemented the repetitive TM measurements, it has been known that, once the TM is characterized at a specific axial position (i.e., on a single plane), optimal input fields for 3D addressable focus can be deduced from the acquired TM, in turn enabling the 3D focus control over a certain 3D range without or with the ‘memory effect’ [M. Jang et al., *Nature Photonics*, 12, 84 (2018), A. Boniface et al, *Optica*, 4, 54 (2017), I. M. Vellekoop et al, *Optics letters*, 8, 1245 (2010)]. In the following, we provide detailed discussions about the two approaches – basis transformation-based and memory effect-based – for implementing 3D addressable focusing without multiple TM measurements and explain the perspective of those approaches for the FLASH focusing technique.

Basis Transformation

In principle, given that a TM is characterized for a large transversal area at a fixed plane $z = z_0$ whose transversal coordinate is set as (x, y) , one may perform a basis transformation on the measured TM to computationally construct a new TM with the output plane $z = z'$ whose transversal coordinate is set as (x', y') . More specifically, the TM for the new plane $z = z'$ can be constructed via $K_{z=z'} = T \times K_{z=z_0}$ where T is the transformation matrix whose column encodes the discretized spherical wavefront information (i.e., $e^{\pm ikr}$ where $r = \sqrt{(x' - x)^2 + (y' - y)^2 + (z' - z_0)^2}$. Note that the amplitude variation in the spherical wavefront is omitted for simplicity.) from the plane $z = z_0$ to the plane $z = z'$. The sign on the exponential term depends on the convergence of the spherical wavefront at the perspective seen from the plane $z = z'$. Once the new TM is properly constructed, one may use a column of the phase-conjugated matrix $K_{z=z'}^*$ to focus at a specific transversal position at the plane $z = z'$. This process can be in effect considered as the reconstruction of an ideal converging or diverging wavefront at the plane of $z = z_0$ which is focused at the plane of $z = z'$.

Here, to ensure the lossless transformation, the original TM needs to be characterized for a large area with a high sampling rate. For instance, it requires characterizing an extremely large transmission matrix with the number of output modes of around 10^8 to address a micrometer-sized focus over a millimeter-scale volume as in our study. This process takes prohibitively long time for a conventional scattering medium so that we opted to measure the transmission matrices multiple times as explained above.

In our previous study, we have demonstrated that the “randomness” of the TM can be specifically designed with a disorder-engineered metasurface platform [Mooseok Jang et al., *Nature Photonics*, 12, 84-90 (2018)]. We anticipate that such approaches can be incorporated with the proposed FLASH technique to get rid of the needs of repetitive TM measurements.

Memory Effect

As an alternative, given that a scattering medium present a sufficiently large memory effect range, one may superpose a spherical wavefront directly on a phase-conjugated field of the measured TM (i.e., a column of $K_{z=z_0}^*$). Specifically, one may take a column $C_{x,y}$ of $K_{z=z_0}^*$ corresponded to a desired focus at (x, y) , then multiply discretized spherical wavefront $e^{\pm ikr}$ (where $r = \sqrt{(x_{\text{input}} - x)^2 + (y_{\text{input}} - y)^2 + (z_{\text{desired}} - z_0)^2}$) to axially translate the focus to the plane of $z = z_{\text{desired}}$. The key difference to the basis transformation strategy is that this strategy

directly manipulates the input wavefront and aims to utilize the intrinsic correlation within a transmission matrix. Therefore, the transmission matrix does not have to be characterized for the large number of output modes.

This correlation effect, often called short-range correlation, only holds within a certain angular range which is inversely proportional to the thickness of a scattering medium L , more specifically, $\Delta\theta_{\text{range}} \sim \lambda/2\pi L$. In practice, the opal glass with a near Lambertian scattering profile we used was experimentally measured to exhibit $\Delta\theta_{\text{range}} < 0.1^\circ$ (see Supplementary Fig. 6 in the revised Supplementary Information), which is too narrow for addressing a millimeter-scale volume with a micrometer-sized focal spot. Moreover, due to our one-dimensional wavefront modulation scheme, this strategy is only applicable to impose a 1-D zone-plate-like pattern along a single direction like a tunable cylindrical lens.

We have clarified how the TM was measured to address focal spots in different axial positions in the Results section of the revised manuscript (Item 9 of List of Changes) and also added Supplementary Text 2 to fully describe the potential approaches to address focal spots in 3-D space without multiple TM measurements (Item 10 & 21 of List of Changes).

R1C6

It might be fruitful to discuss on using a pulsed laser as the light source. As the authors suggest, one can use multiple lines for wavefront shaping to obtain a high-quality focus. For a CW laser, the power is simply wasted in the transiting phase as the signal in the transition phase would be a mix of two independent foci. This would be an issue since the efficiency of FLASH is considerably low. Synchronizing the pulsed laser with the scanning, one can minimize the loss.

We deeply appreciate for the insightful comment. We agree that synchronizing a pulsed laser with a line-beam scanning can be an effective solution to avoid an undesired interference effect (detailed in R1C4) and minimize an energy loss during the transition phase in the FLASH focusing technique. To fully exploit the MHz-modulation speed of the FLASH without energy loss, it would be desired to synchronize the pulsed laser source of the pulse repetition rate higher than 1 MHz (for instance, acousto-optic modulator (AOM)-based Q-switched lasers) with a line beam scanning unit. We have added a brief discussion on the potential use of a pulsed laser in the Discussion section of the revised manuscript (Item 11 of List of Changes) and fully described about the optimal implementation of the FLASH technique, including the use of a polygon mirror, a F-theta lens, and a pulsed laser, in the Supplementary Text 1 (Item 1 of List of Changes).

R1C7

I find supplementary figure 3b a bit confusing. I am not sure if the position in scan direction refers to the displacement of the generated focus. In principle, the focus should be independently addressable and there is no correspondence between the direction of the scan of the direction of the focus' shift. When I saw the labels sitting above the figure, I thought this was the actual dimension of this figure. However, this interpretation is wrong as the scale bar is way smaller than that. Clarification is needed for this figure.

Thank you for the careful reading. We have revised the corresponding labels and the figure caption in Supplementary Figs. 3b and 3c to clarify that the presented figures are about the line-beam intensity profiles on the DMD surface (Item 12 of List of Changes).

R1C8

I sometime misread the inset frequency response figure (such as Fig. 4c) as the zoomed version of the time sequence. I think choosing a different line style (or simply changing the color) may help for a clearer presentation.

Thank you for the suggestion. To avoid the confusion, we have chosen to use the dotted line style for the plots presenting the frequency responses in Fig. 4c, Fig. 5e, and Supplementary Figs. 11a-11b (Item 13 of List of Changes).

R1C9

It might be better to write the factor for binary phase modulation as $1/(2\pi)$ or $(2\pi)^{-1}$ to avoid ambiguity. This applies to other numbers in the manuscript. But I think this can be corrected in the copyediting.

Thank you for the thoughtful comment. To avoid this ambiguity, we have made revisions throughout the main text to include parenthesis in the denominator (Item 14 of List of Changes).

R1C10

Overall, I think this is an interesting work for high-speed, random-access focus. The flexibility tuning between high refreshing rate and the quality of the focus. It can be a good complement to existing random-access techniques once my concerns are addressed.

We sincerely appreciate the Reviewer for the constructive and encouraging comments. It indeed helped us a lot getting a deeper understanding on the proposed technique and improving the manuscript.

Reviewer #2 (Remarks to the Author):

In this manuscript, the authors detail a scheme ("FLASH") for fast focus shifting and deflection at ~10 MHz rates, approximately one order of magnitude faster than currently reported state of the art (e.g. galvanometer scanners). The authors then provide a very robust demonstration of their scheme demonstrating dynamic on-demand focal shifting at millimeter scale volumes. The experimental documentation both in the manuscript and supplementary materials is commendably thorough.

We thank the Reviewer for the positive comments and the recommendation for publication.

R2C1

While the idea of spatially/temporally multiplexing a DMD/SLM with a cylindrical lens and actuator of some form is not particularly new (e.g. [1,2]), the authors achieve a commendably high sampling rate and then combine this with a scattered media focusing element for dynamic focal scanning in a volume. The combination of spatiotemporal multiplexed SLM with a scattering medium as a holographic focusing element at high speeds is novel and interesting, and I would recommend publication.

[1] Teng, D., Liu, L., Zhang, Y., Pang, Z., Chang, S., Zhang, J. and Wang, B., 2014. Spatiotemporal multiplexing for holographic display with multiple planar aligned spatial-light-modulators. *Optics Express*, 22(13), pp.15791-15803.

[2] Takaki, Y. and Okada, N., 2009. Hologram generation by horizontal scanning of a high-speed spatial light modulator. *Applied Optics*, 48(17), pp.3255-3260.

Thank you for bringing the previous works to our attention. We agree that those previous works are highly relevant to our work. They used a multiplexing scheme comprised of the similar components to ours to demonstrate the modulation capability that is conventionally unachievable. Here, we have described the relevance between those earlier works and ours, and explained why the gain in temporal DOF is the unique feature of the proposed FLASH technique (i.e., the key distinction between the earlier works and ours).

The primary goal of the previous works is to implement a 3D holographic display system with a large view angle. In other words, the time-multiplexing technique is used to implement larger spatial degrees of freedom (DOF), not larger temporal DOF (i.e., higher modulation speed) as in ours. More specifically, in the earlier works, many 'elemental' holograms that are sequentially displayed with 2D-SLMs are spatially stitched to construct a single large 'complete' hologram by scanning a galvanometer (Reviewer's Ref. [1]) or an actuated cylindrical lens (Reviewer's Ref. [2]). Therefore, with the scanning frequency of f_{scan} and the SLM's intrinsic refresh rate of f_{slm} , the 'complete' hologram consists of f_{slm}/f_{scan} elemental holograms (i.e., the spatial degrees of freedom was improved by the factor of f_{slm}/f_{scan}) while the modulation speed is reduced by the same factor (i.e., the temporal degrees of freedom is reduced by the factor of f_{slm}/f_{scan}), with the setting $f_{scan} < f_{slm}$. For instance, in the Reviewer's Ref. [2], f_{slm} and f_{scan} was set to 13.33 kHz and 60 Hz so that the spatial DOF was increased by around 200 times but the temporal DOF was reduced by around 200 times compared to the intrinsic refresh rate of the DMD, f_{slm} . We note that the word 'space-multiplexing' was used in the Reviewer's Ref. [1] to indicate the use of multiple SLMs to generate a single hologram, not to indicate 'space-multiplexing' in a time domain

as in ours. In this class of techniques, the effective modulation speed is always lower than or same with f_{slm} .

In the FLASH technique, the multiplexing method is exploited in the exactly reversed way. More specifically, the spatial multiplexing technique is used to implement larger temporal DOF (i.e., higher modulation speed) beyond the intrinsic modulation speed of SLMs, f_{slm} . For the comparison sake, in our experiments, the temporal DOF (i.e. modulation speed) was increased by around 1,000 times while the spatial DOF was reduced by around 1,000 times.

We have clarified the relevance and the key difference between the previous efforts and our approach in the Principle section of the revised manuscript with the citations (Item 15 & 20 of List of Changes).

R2C2

I do, however, have one comment for the authors pertaining to the discussion section. The claim was made that FLASH is suitable for "non-mechanical wide-angle beam steering for LiDAR", but to me this claim seems problematic. Generally, the performance of beam-steering systems can be characterized by the steering angular range, angular resolution, and number of resolvable beam directions, beam throughput, and speed [3-5]. Angular range, angular resolution, and number of resolvable directions are, amongst other things, a function of the number of elements within the beam steering system and the spacing of those elements [3-5]. The approach taken by the authors with FLASH is to take an $M \times N$ DMD and sample its rows one by one each duty cycle. This achieves a large gain in sampling rate, but also reducing the instantaneous number of elements available for beam steering to N . N elements is less than $M \times N$ elements of the standard DMD, and so there will a degradation in some of angular range, angular resolution, and number of resolvable directions. I think that if the authors want to include "wide angle beam steering" as a potential application, including a discussion on the trade-offs of FLASH with respect to the standard DMD in terms of angular range and resolution would be warranted.

[3] Xu, J., Cua, M., Zhou, E.H., Horie, Y., Faraon, A. and Yang, C., 2018. Wide-angular-range and high-resolution beam steering by a metasurface-coupled phased array. *Optics Letters*, 43(21), pp.5255-5258.

[4] Guo, Y., Guo, Y., Li, C., Zhang, H., Zhou, X. and Zhang, L., 2021. Integrated optical phased arrays for beam forming and steering. *Applied Sciences*, 11(9), p.4017.

[5] Heck, M.J., 2017. Highly integrated optical phased arrays: photonic integrated circuits for optical beam shaping and beam steering. *Nanophotonics*, 6(1), pp.93-107.

We thank the Reviewer for the insightful comments regarding the beam steering application. As the Reviewer pointed out, when a DMD (i.e., SLM) used without incorporating a scattering medium, the 2D angular range θ_{rng} and the 2D angular resolution $\delta\theta$ obeys a tradeoff relationship, which is described as $\theta_{rng}/\delta\theta = N_{dir} = DOF_{spatial}$ where N_{dir} and $DOF_{spatial}$ are the number resolvable beam directions and the number of independently controllable elements of the DMD, respectively. Therefore, N_{dir} is limited to $DOF_{spatial}$.

In contrast, when a scattering medium is incorporated, the tradeoff relationship between θ and $\delta\theta$ can be circumvented, leading to an extremely large N_{dir} as demonstrated in the previous work based on the same interferometric focusing scheme (Reviewer's Ref. [3]). In this scheme, the angular range θ_{rng} and the angular resolution $\delta\theta$ can be arbitrarily set by choosing a proper scattering medium and a projection area. For instance, in the Reviewer's Ref. [3], it

has been shown that the angular range θ_{rng} can be extended up to $\pm 80^\circ$ with the angular resolution $\delta\theta$ of about 0.01° for both axes, resulting in the number of resolvable directions of 5×10^7 ($= N_{\text{dir}}$) which is an order of magnitude larger than the $\text{DOF}_{\text{spatial}}$ of the DMD ($= M \times N = 1,920 \times 1,080$ in the range of 10^6).

However, the performance in a scattering-assisted beam-steering is compromised in other aspects – the beam throughput (i.e., fraction of output beam energy in the desired direction). With the ideal scattering medium with the near-unity transmission, the beam throughput to a desired direction before optimization is given as $1/N_{\text{dir}}$ so that the beam throughput after wavefront optimization is estimated to $\text{DOF}_{\text{spatial}}/N_{\text{dir}}$ with the approximated enhancement factor $\text{DOF}_{\text{spatial}}$. In other words, to extend N_{dir} beyond $\text{DOF}_{\text{spatial}}$, the beam throughput has to be proportionally compromised. In practice, a scattering medium has a low transmittance, resulting in the further reduction in the beam throughput. For instance, in the Reviewer's Ref. [3], the measured beam throughput was around 10^{-6} . The full description on the relation between the spatial DOF, the number of resolvable angles, and the beam throughput for the wide-angle beam steering application may be found in the Reviewer's Ref. [3].

Additionally, in our sequential 1-D modulation scheme, $\text{DOF}_{\text{spatial}}$ is reduced from $M \times N$ to N to increase $\text{DOF}_{\text{temporal}}$ by the factor of M so that there will be an additional penalty on the beam throughput when the FLASH technique is used for wide angle beam steering applications. Nonetheless, for certain wide beam steering applications, especially with a high-power laser, we believe that our method can be still an effective option as it provides superior characteristics of wide angular range and ultrafast steering speed. As suggested by the Reviewer, we have clarified this limitation in the Discussion session of the revised manuscript with the citation (Item 3 & 20 of List of Changes).

Reviewer #3 (Remarks to the Author):

In this manuscript the authors propose and demonstrate a method to quickly generate a focus at a desired position. The basic idea is that light passing through a scattering medium can be brought into a focus if the incident wavefront has been suitably shaped (ref 29). If we want a focus at a different position we need a different incident pattern (in this case the authors used binary amplitude modulation, which has been shown to work surprisingly well in ref 35), so creating a focus at a desired position on demand can be reduced to select one of the pre-obtained patterns. The authors achieve this by having each pattern as a single column on a DMD, and then scanning to select the desired one.

I don't have much of a problem with the method proposed or with the experiments presented. They do a satisfactory job at supporting the claims.

We thank the reviewer for the concise description and the positive comments. In the light of the Reviewer's comment, we have strengthened the manuscript with the in-depth discussions and details regarding our wavefront shaping experiments.

R3C1

My only real problem with the paper is the absolute lack of any discussion on the "wavefront shaping" part of the experiment. Light is assumed to form a focus, and the details and limitations of this approach are never discussed. For instance I didn't manage to find any information about the scattering medium itself. What is its mean free path? How stable is it over time? Also the fact that this approach will intrinsically yield small "enhancement factors" (i.e., ever focus will always be surrounded by a visible speckle pattern) is just mentioned in passing.

In conclusion, this is an overall nice paper, but I really wish the authors discussed more in depth the wavefront shaping part of the experiment.

We deeply appreciate the Reviewer pointing out those important omissions. In this revision, we have thoroughly gone through the manuscript to supplement the following details and in-depth discussions regarding the wavefront shaping part of our experiment.

Detailed information on a scattering medium

The scattering medium used as a holographic focusing element was an opal diffusing glass from Edmund (#46-167) with a near Lambertian scattering profile whose thickness L and transmittance T are 0.45 mm and 0.31. For a such thick diffusive medium, the ballistic component through the medium is too small to reliably estimate a scattering mean free path l_s experimentally, using the Beer-Lambert law. The transport mean free path l_t is more relevant information in our experimental context, which has been estimated to be 100 μm in the previous study [I Freund and M Rosenbluh, "Memory Effects in Propagation of Optical Waves through Disordered Media", *Physical Review Letters* 61 2328 (2021)]. To further supplement the wavefront shaping-related characteristics of the scattering medium, we have performed additional experiments to measure (1) the stability of the medium over time and (2) the angular memory effect range of the medium, respectively. In the following, we describe the experimental details and results in details.

(1) Stability of scattering medium over time: During the period of measuring a transmission matrix (TM) and implementing applications based on the measured TM (e.g., scanning a focal spot over 3D volume), the scattering medium needs to be stable (i.e., the TM of the medium needs to be unchanged). To clarify that our experimental condition meets the above requirement, we have measured how stable the medium is over time as the Reviewer suggested. In this experiment, a laser beam with a diameter of 4 mm was projected onto the scattering medium that was rigidly clamped with a filter mount (Thorlabs, DH1/M), and the speckle patterns behind the medium at the target plane of $z = 2$ mm were repeatedly captured with a camera over a time period of ~ 40 minutes. In **Fig. R3** below, the correlation coefficients between the first speckle pattern and the subsequent speckle patterns were plotted as a function of time, and the error bar shows the standard deviation of five measurements. With the definition of the stability time as the time when the correlation coefficient drops to 0.5, we could determine the stability time to roughly 30 minutes. In contrast, in our experiment of measuring the TM whose columns N are 352, we typically projected ~ 2000 input patterns onto the scattering medium by updating a single DMD column at a refresh rate of ~ 1 kHz and so the overall time required for the TM measurement was approximately 2 seconds. Considering the stability time of ~ 30 minutes, we can say that the TM measurement can be almost instantaneously completed. Furthermore, we note that this stability time highly depends on the mechanical environment such as the mechanical fluctuation of optical table and the clamping method for an opal diffuser.

Figure R3 | Correlation coefficient between the first speckle pattern and subsequent speckle patterns as a function of time for an opal diffusing glass.

(2) Angular range of memory effect of scattering medium: When an incident beam is tilted within a certain angular range, the resulting speckle pattern becomes identical to the speckle pattern with a normally incident beam except that it will be spatially shifted at the projection plane behind a scattering medium. This effect is called angular memory effect. In many previous studies [*I. M. Vellekoop et al, Optics letters, 8, 1245 (2010)*, *O. Katz et al, Optica, 1, 170 (2014)*], this memory effect have been exploited to translate the holographic focus behind a scattering medium by directly superposing a spherical wavefront or a tilted

wavefront on a phase conjugated wavefront. Here, to see whether the memory effect of our scattering medium is practically useful in translating a focus in 3-D space, we have measured an angular memory effect range.

In this experiment, a laser beam with a diameter of 5 mm illuminated the fixed area of the scattering medium, and then the speckle pattern was captured by a camera at a plane 70 mm behind the medium (**Fig. R4a**). In **Fig. R4b** below, the correlation coefficients between the first speckle pattern at $\delta\theta = 0^\circ$ and the subsequent speckle patterns at each $\delta\theta$ were computed as a function of the rotation angle $\delta\theta$ of the medium, and the error bar indicates the standard deviation of five measurements. When defining the angular range where the correlation coefficient drops to 0.5 as the memory effect range $\Delta\theta_{memory}$, we could estimate it to be about 0.035 degrees. Based on this memory effect range, given the focal length of 2 mm (i.e., the target plane is set to 2 mm behind the medium), the lateral range over which a focal spot can be shifted is only around $1\ \mu\text{m} \times 1\ \mu\text{m}$, which is much narrower than the 3D addressable range ($5\ \text{mm} \times 5\ \text{mm} \times 5.5\ \text{mm}$) demonstrated in our study. We anticipate that the use of a disorder-engineered metasurface with a wide memory effect range would be an effective alternative in this context.

Figure R4 | Measurement of angular range of memory effect. **a**, Schematic of the optical setup to measure the angular correlation range of an opaque diffusing glass as a scattering medium. **b**, Measured memory effect range for the opaque diffusing glass.

Focusing scheme based on the measurement of TM

To form a focal spot behind the scattering medium with the FLASH technique, we took the following procedure; (1) measuring TM of Hadamard basis, (2) transforming the TM from 'Hadamard' basis to 'position' basis, (3) computing conjugated transpose of the TM, (4) computing a binarized amplitude pattern of the phase conjugated field for making a focal spot at a desired position, (5) displaying that binarized pattern on a DMD column. In the original manuscript, we have described the above-mentioned procedure in the fourth paragraph of

Result section of 'Validation of the FLASH focusing technique'. We have further augmented this description of the procedure to describe how the spot is formed (Item 16 of List of Changes). Also, in the revised manuscript and Supplementary Information, we have described how we address focal spots in different axial positions (Item 9 of List of Changes) and provided the alternative wavefront shaping techniques - 'basis transformation-based' and 'memory effect-based' - to address focal spots in a 3-D space without repetitive measurement of TMs (Item 10 of List of Changes).

Small enhancement factor (i.e., Small PBR)

As the Reviewer pointed out, the small enhancement factor is an important limitation of our proposed approach. We have modified the Discussion section of the main text to explicitly discuss about this limitation and its implications in practical applications (Item 3 of List of Changes). In addition, we have added Supplementary Text 4 to provide our full consideration on practical applications of FLASH technique in relation to the low enhancement factor (Item 4 of List of Changes).

We summarize our revision on the main text and the Supplementary Information regarding in-depth discussion and details of the wavefront shaping part:

1. Providing the optical properties – material, thickness, transmittance - of the scattering medium
(Item 17 of List of Changes)
2. Performing additional experiments to measure the stability and the memory effect range of the scattering medium
(Item 18, 19 of List of Changes)
3. Augmenting our description on how the spot is formed through the process of TM measurement and phase conjugation
(Item 16 of List of Changes)
4. Providing experimental details on addressing focal spots in different axial positions & providing alternative wavefront shaping approaches
(Item 9, 10 of List of Changes)
5. Discussing the limitations associated with the small enhancement factor & providing the perspectives on practical applications of FLASH technique in relation to the small enhancement factor
(Item 3, 4 of List of Changes)
6. Performing additional experiments on scanning fluorescence imaging to see the effect of small enhancement factor
(Item 5 of List of Changes)
7. Providing detailed discussion on the experimental configuration for optimal wavefront shaping performance
(Item 1 of List of Changes)
8. Discussing the fundamental stochastic nature of a scattering medium and its implication – statistical peak intensity fluctuation - in FLASH focusing
(Item 8 of List of Changes)

REVIEWERS' COMMENTS

Reviewer #1 (Remarks to the Author):

In the revision, the authors addressed most of my questions. They demonstrated what FLASH could potentially achieve in practice. Theoretical and simulation results on the random fluctuations of the focal spot intensity are solid and clearly explained why we saw fluctuations in their experiment. Issues with respect to the presentation clarity are properly addressed. This work presented a versatile controlling scheme in wavefront shaping in offering the freedom in balancing between a high refreshing rate and the quality of the focus. Thus, I would like to recommend this work for publication.

Some minor comments with respect to the revision:

1. The widefield result for the fluorescence beads shown in Supplementary figure 12 looks like a defocused image. If that's the case, I would recommend the authors re-image the sample to acquire a proper ground truth.
2. Some of the texts were cropped in Supplementary figure 2 (upper left corner). The authors may want to handle this cropping issue in their final version.

In summary, the authors did a thorough and solid investigation on FLASH and I believe it will be a valuable technique in the wavefront shaping field.

Reviewer #3 (Remarks to the Author):

The authors' rebuttal is EXTREMELY lengthy and detailed. Which in principle is a good thing, but in practice hides the fact that most of the changes have been made to the supplementary information, and not to the main text of the manuscript. Furthermore, some changes in the main text (e.g. the replies labelled as item 3 in the provided list), are both an acknowledgement of a problem noticed by a reviewer (in this case: low enhancement factor inherent to the method), and a claim that those problems are not important. I feel that a less hype-prone discussion would make for a better manuscript.

Reviewer #1 (Remarks to the Author):

In the revision, the authors addressed most of my questions. They demonstrated what FLASH could potentially achieve in practice. Theoretical and simulation results on the random fluctuations of the focal spot intensity are solid and clearly explained why we saw fluctuations in their experiment. Issues with respect to the presentation clarity are properly addressed. This work presented a versatile controlling scheme in wavefront shaping in offering the freedom in balancing between a high refreshing rate and the quality of the focus. Thus, I would like to recommend this work for publication.

We thank the Reviewer for acknowledging the key changes in our revised manuscript and for the recommendation of the publication.

R1C1

The widefield result for the fluorescence beads shown in Supplementary figure 12 looks like a defocused image. If that's the case, I would recommend the authors re-image the sample to acquire a proper ground truth.

We appreciate the careful examination. We confirm that the sample position was aligned along the optical axis as carefully as possible to acquire the widefield image in Supplementary Figure 12. As evidenced in Figure R1 below, the size of beads estimated from the 1D intensity profiles matches the nominal size of the beads ($\sim 1 \mu\text{m}$) indicated in the catalog (Invitrogen, F13082). Because a conventional bright-field imaging configuration lacks of depth-sectioning capability, the imaging result presented in Supplementary Figure 12g can present some blurring effect compared to the imaging results in Supplementary Figure 12c-f acquired in a confocal imaging configuration.

Figure R1 | **a**, Widefield image of fluorescence beads shown in Supplementary Fig. 12g. Scale bar: $1 \mu\text{m}$. **b**, Intensity profiles of three isolated beads along red lines shown in **a**.

R1C2

Some of the texts were cropped in Supplementary figure 2 (upper left corner). The authors may want to handle this cropping issue in their final version.

We have resolved the clipping issue from PDF conversion to prepare the final version of the Supplementary Information.

Concluding remarks from Reviewer #1

In summary, the authors did a thorough and solid investigation on FLASH and I believe it will be a valuable technique in the wavefront shaping field.

Reviewer #3 (Remarks to the Author):

The authors' rebuttal is EXTREMELY lengthy and detailed. Which in principle is a good thing, but in practice hides the fact that most of the changes have been made to the supplementary information, and not to the main text of the manuscript. Furthermore, some changes in the main text (e.g. the replies labelled as item 3 in the provided list), are both an acknowledgement of a problem noticed by a reviewer (in this case: low enhancement factor inherent to the method), and a claim that those problems are not important. I feel that a less hype-prone discussion would make for a better manuscript.

We respectfully disagree with the Reviewer's opinion. In the previous revision, we have carefully addressed the potential effect of low enhancement factor and specifically described about the limitation with a full paragraph in the Discussion session (as our response to the comments from the Reviewer #1 and #3) as follows:

“However, unlike conventional lenses, the scattering-assisted lens does not directly form the image of the entire 2D plane at a desired depth, preventing their use in single-shot imaging applications such as bright-field microscopy. In addition, in contrast to conventional varifocal lenses, the scattering-assisted holographic focusing is typically associated with background intensity fluctuations and lower transmittance. Because the FLASH technique increases the modulation speed at the cost of $\text{DOF}_{\text{spatial}}$, it intrinsically yields a small focal contrast (i.e., the peak-to-background ratio), which would be a limiting factor in imaging applications (see Supplementary Text 4 for the implication of the small focal contrast). More specifically, the aggregated background signals from the speckle granules around a focal spot deteriorates the image contrast (see Supplementary Text 5 and Supplementary Fig. 12 for the demonstration of scanning fluorescence imaging based on the FLASH technique).”

We think that the main manuscript is well organized to deliver the major contribution of our study - (1) presenting a versatile wavefront shaping scheme to reallocate the DOFs in spatiotemporal domain, (2) demonstrating that the proposed scheme can bypass the limitation in modulation speed with excessive spatial DOFs, and (3) validating the record-high modulation speed (~10 MHz) in controlling optical focus in a large 3D space.